# Bridging Intuition and Data: A Unified Bayesian Framework for Optimizing Unmanned Aerial Vehicle Swarm Performance

**DOI:** 10.3390/e27090897

**Published:** 2025-08-25

**Authors:** Ruiguo Zhong, Zidong Wang, Hao Wang, Yanghui Jin, Shuangxia Bai, Xiaoguang Gao

**Affiliations:** 1School of Electronics and Information, Northwestern Polytechnical University, Xi’an 710129, China; zhongruiguo@mail.nwpu.edu.cn; 2Thrust of Intelligent Transportation, Hong Kong University of Science and Technology (Guangzhou), Guangzhou 511455, China; 3Department of Computer Science, City University of Hong Kong, Hong Kong, China; zidowang@cityu.edu.hk (Z.W.); sx.bai@my.cityu.edu.hk (S.B.); 4Xi’an Electronic Engineering Research Institute, Xi’an 710100, China; wh1998@mail.nwpu.edu.cn; 5Nanjing Research Institute of Electronic Technology, Nanjing 210039, China; jyh960612@163.com

**Keywords:** low-altitude economy, UAV swarm, Bayesian Network (BN), variance decomposition, Multicriteria Decision-Making (MCDM)

## Abstract

The swift growth of the low-altitude economic ecosystem and Unmanned Aerial Vehicle (UAV) swarm applications across diverse sectors presents significant challenges for engineering managers in terms of effective performance evaluation and operational optimization. Traditional evaluation methods often struggle with the inherent complexities, dynamic nature, and multi-faceted performance criteria of UAV swarms. This study introduces a novel Bayesian Network (BN)-based multicriteria decision-making framework that systematically integrates expert intuition with real-time data. By employing variance decomposition, the framework establishes theoretically grounded, bidirectional mapping between expert-assigned weights and the network’s probabilistic parameters, creating a unified model of subjective expertise and objective data. Comprehensive validation demonstrates the framework’s efficacy in identifying critical performance drivers, including environmental awareness, communication ability, and a collaborative decision. Ultimately, our work provides engineering managers with a transparent and adaptive tool, offering actionable insights to inform resource allocation, guide technology adoption, and enhance the overall operational effectiveness of complex UAV swarm systems.

## 1. Introduction

The rapid expansion of the low-altitude economy has intensified the deployment of Unmanned Aerial Vehicle (UAV) swarms across diverse sectors such as logistics Du et al. [1], agriculture Volpato et al. [2], and disaster response Medeiros et al. [3], Calamoneri et al. [4]. While UAV swarms promise enhanced operational efficiency, expanded coverage, and improved resilience Du et al. [5], their effective management creates significant challenges for engineering managers, who must evaluate overall performance, optimize resource allocation, and make strategically sound decisions in complex, dynamic environments.

To tackle these challenges, some research has explored various facets of UAV adoption and operational hurdles. For instance, Ali et al. [6] have focused on organizational readiness for UAV uptake and identified implementation barriers in specialized sectors like rural healthcare Koshta et al. [7]. While valuable for initial adoption strategies, this work offers limited guidance for managing the performance of already deployed swarms. Other approaches have used advanced models to evaluate the performance of individual UAVs Hossain et al. [8]. However, these individual assessments are insufficient as they typically overlook the emergent behaviors and complex interdependencies inherent in swarm dynamics, leaving a critical research gap at the system-level of analysis.

From an Operational Research (OR) perspective, the selection of an optimal swarm configuration, operational doctrine, or enabling technology constitutes a classic, yet highly complex, multicriteria decision-making problem. For instance, a logistics manager must decide whether to prioritize payload capacity for efficiency or invest in enhanced detection sensors for safety, especially in dense urban environments Hossain et al. [8]. Similarly, a disaster response coordinator must weigh the need for rapid deployment against the importance of robust, interference-resistant communication links Yan et al. [9]. These decisions are not static; the optimal choice depends on the specific mission profile and the operating environment.

Moreover, considering expert knowledge alone is insufficient for this task. An expert’s judgment is a static snapshot in time and can be prone to cognitive biases, struggling to account for the emergent, nonlinear behaviors of a complex swarm or adapt to rapid changes in technology and operational conditions. Therefore, a robust framework must be data-driven, capable of integrating real-world performance data to uncover hidden bottlenecks, validate or correct expert intuition, and dynamically adapt its evaluation criteria over time. It is precisely this need for data-driven adaptability that renders traditional, static Multicriteria Decision-Making (MCDM) frameworks like the Analytic Hierarchy Process (AHP) inadequate for the task.

Conversely, while modern neural network-based methods are designed for data-driven adaptation Wang et al. [10], Guo et al. [11], their opaque, “black-box” nature lacks the interpretability required for high-stakes managerial oversight. In safety-critical engineering applications, this lack of transparency is a significant liability, as it erodes trust and hinders a manager’s ability to make accountable, auditable decisions. This creates a crucial methodological gap for a framework that is simultaneously interpretable by design, adaptive to new data, and capable of systematically integrating expert judgment.

To address this critical gap, this paper introduces a novel Bayesian network (BN)-based multicriteria decision-making framework. BNs are uniquely suited for this task because their graphical structure provides a transparent representation of complex dependencies, while their probabilistic nature is ideal for modeling uncertainty and learning from data [12]. Our framework’s primary contribution is a theoretically grounded, bidirectional mapping between expert-derived weights and the network’s probabilistic parameters, achieved through variance decomposition. This dual methodology creates a seamless fusion of subjective expertise and objective data, empowering managers to identify critical performance drivers, make data-informed decisions, and dynamically adapt strategies to evolving mission requirements.

The key contributions of this study are as follows:

**Unified Framework for Data and Intuition:** By employing CPTs within BN, the proposed framework effectively integrates real-time data with expert judgment, providing a consistent and unified representation of diverse information sources.

**Adaptive and Dynamic Decision Making**: The framework’s ability to dynamically update priority weights based on real-time data and expert inputs ensures robust decision-making in the face of changing mission conditions and varying data volumes.

**Systematic Application in UAV Swarm Management**: The research presents a structured methodology for establishing a hierarchical evaluation network for UAV swarms and for determining the relative importance of various performance factors. This provides comprehensive and actionable guidance for managers involved in the organization, deployment, and iterative improvement of UAV swarm systems.

**Applicability to Diverse Scenarios**: The framework is validated through comprehensive UAV swarm performance evaluation scenarios, showcasing its potential utility for application across various sectors.

The remainder of this paper is structured as follows: Section 2 reviews the relevant literature. Section 3 describes the proposed BN-based framework and its components. Section 4 presents the comprehensive experimental validation. Section 5 discusses the key findings, implications, and limitations of this study. Finally, Section 6 concludes the paper.

## 2. Literature Reviews

In this section, we review literature pertinent to the core challenges of this study. (1) We examine research developments in UAV swarm technology, as shown in Figure 1, focusing on aspects that present significant operational management and decision-making challenges for engineering managers. (2) We critically assess existing MCDM frameworks, particularly concerning their suitability for integrating the diverse data sources and expert knowledge essential for effective UAV swarm performance evaluation from a managerial view.

### 2.1. Studies on UAV Swarm

UAV swarms have attracted substantial attention because they enable complex collaborative tasks that are difficult or impossible for a single, large UAV. By distributing responsibilities among multiple smaller UAVs, swarms gain advantages in redundancy, flexibility, and scalability. However, effectively managing a swarm involves challenges in mobility, environmental detection, communication and decision-making Kong et al. [13].

**Swarm Mobility:** Research on swarm mobility has largely focused on path planning and formation control strategies to ensure collision-free and energy-efficient navigation Pan et al. [14]. Typical approaches optimize flight trajectories by accounting for constraints such as wind, battery performance, and communication quality. Advanced formation control methods further enhance coordinated tasks like search-and-rescue in complex terrains. From an engineering management perspective, optimizing these mobility strategies involves complex trade-offs between mission effectiveness (e.g., coverage, speed) and operational constraints (e.g., energy consumption, airspace regulations, fleet size), requiring robust evaluative frameworks to support strategic and tactical decisions.

**Swarm Environmental Detection:** Collaborative sensing enables UAVs to detect and share environmental information in real time, improving both individual and collective situational awareness Zhou et al. [15]. Communication algorithms must not only handle bandwidth allocation and data routing but also remain robust under interference or partial communication failures Chen et al. [16]. The effective management of these sensing and communication resources is a critical concern for engineering managers, as it directly impacts the reliability of shared situational awareness, the quality of collective decision making, and ultimately, mission success rates under varying operational conditions, as shown in Figure 2.

**Swarm Decision Control:** High-stakes missions require real-time decision making which entails predicting target trajectories, monitoring energy consumption, and tracking sensor health Yao et al. [17]. With robust communication in place, these processes can be executed collectively by the swarm—either in a decentralized or semi-centralized manner—affording the agility needed to adapt to rapidly changing environments. For engineering managers, ensuring that the swarm’s decision control architecture aligns with mission objectives, risk tolerance, and human oversight capabilities is paramount. This involves evaluating the effectiveness of different control paradigms (centralized, decentralized, hybrid) in terms of responsiveness, scalability, and resilience.

**Mission Capability:** Determining optimal UAV assignments is central to minimizing costs and maximizing the benefits of swarm deployment. Scholars have investigated algorithms that adapt to shifting mission objectives—ranging from medical deliveries in disaster zones to urban logistics Banik et al. [18], Wu et al. [19]. Recent work has highlighted hybrid strategies that merge centralized planning (for rapid, coordinated task allocation) with decentralized, bio-inspired labor division Peng et al. [20] or auction-based protocols Wang et al. [21], demonstrating the potential for robust, scalable solutions under evolving operational demands. For an engineering manager, the challenge lies not only in selecting the most appropriate task allocation strategy for a given mission profile but also in continuously evaluating its effectiveness against key performance indicators (KPIs) like completion time, resource utilization, and adaptability to unforeseen events. This requires an evaluative framework that can assess the performance of these complex, dynamic allocation systems.

**Swarm Viability:** Swarm resilience and anti-interference strategies serve as the foundations for stable operations in hostile or unpredictable environments. Disruptions such as signal jamming, sensor malfunctions, or partial node failures can critically endanger mission success Liu et al. [22]. In response, researchers have explored a range of robustness techniques—from redundant communication channels to advanced error-correction algorithms—to help preserve operational continuity. For engineering managers, decisions about which viability-enhancing technologies to invest in involve a critical cost–benefit analysis, weighing the expense of implementation against the mitigated risk of mission failure. Therefore, a structured methodology to evaluate the effectiveness and relative importance of different resilience factors is essential for making justifiable investments and ensuring the overall viability and sustainability of swarm operations.

Collectively, these studies underscore the technical challenges of UAV swarm systems and propose numerous algorithmic solutions to address them. As UAV missions diversify—from on-demand delivery to disaster management—researchers are expanding performance criteria beyond flight endurance and payload capacity to include collective sensing, real-time responsiveness, and strategic oversight Koshta et al. [7], Hossain et al. [8]. Real-world deployment requires a more comprehensive view. For instance, Dabić et al. [23] analyzes consumer behavior in adopting drone delivery services, showing that perceived ease of use and usefulness encourage uptake, while privacy concerns deter it. Likewise, Banik et al. [18] employ graph theory and matrix approaches to identify the optimal UAV for delivering medical supplies under varied scenarios. In disaster relief, UAVs help minimize human exposure to high-risk areas but demand meticulous planning, inter-agency coordination, and adaptability Yan et al. [9]. Accordingly, effective UAV swarm deployment extends beyond technical innovation to include strategic resource allocation, adherence to policies, and active stakeholder engagement.

Despite ongoing advancements in these technical domains, from an engineering management perspective, a significant gap persists: the lack of a comprehensive and adaptive evaluation methodology that fully integrates these diverse technical performance aspects with real-world operational constraints and strategic objectives. Engineering managers are often left without tools to holistically assess overall swarm readiness, identify critical performance bottlenecks across interdependent functions, or dynamically adjust strategies based on evolving mission data and expert insights. Furthermore, UAV swarm operations inherently generate both quantitative data (e.g., sensor outputs, battery states) and rely on qualitative judgments (e.g., expert operational experience). This underscores the pressing need for decision-support frameworks capable of systematically fusing these heterogeneous information streams. Crucially, for managerial acceptance and effective implementation, especially when expert judgment is a primary input or when outcomes need clear justification, such frameworks must not only be robust but also interpretable, matching or exceeding the performance of existing, often less integrated, approaches.

### 2.2. Studies on Combined Data and Intuition Decision Making

A wide variety of MCDM methods have been developed, each offering advantages under specific conditions Alinezhad et al. [24]. The analytic hierarchy process (AHP) Saaty [25], Bae et al. [26], which involves the hierarchical decomposition of a problem, construction of a criterion hierarchy, and transformation of individual judgments of relative importance into weights, is the most popular method. TOPSIS Boran et al. [27] is another prominent MCDM method widely applied in different fields due to its simplicity and the ability to maintain consistent steps regardless of the problem size. VIKOR Opricovic and Tzeng [28] determines a compromise solution that is closest to the ideal solution based on the adopted distance measures.

While these foundational MCDM methods offer structured approaches for prioritizing criteria and alternatives, their traditional implementations often rely on static pairwise comparisons and predefined weights. This can be a significant limitation for engineering managers overseeing dynamic UAV swarm operations, where mission parameters, environmental conditions, and even the swarm’s capabilities can change rapidly, necessitating evaluation frameworks that can adapt to real-time data and evolving expert understanding.

Predominantly in MCDM approaches, criteria weights are established by subjective methods that represent the decision-maker’s intuition or subjective judgment. Thus, the rankings of alternatives based on weights can be impacted by the decision-maker’s lack of information or experience. Objective weighting techniques frequently employ mathematical models to compute weights but ignore the subjective judgment; these include PCA Abdi and Williams [29] and entropy weighting Zhu et al. [30]. Therefore, a combined weight approach is more acceptable as it uses the strengths of both methods, indicating not only how significant a criterion is to the decision maker but also how diverse the criterion values are across different options.

Hybrid methods have been proposed to address the issue of varying expert preferences. These methods aggregate the individual outputs of various MCDM techniques to obtain a more consistent and reliable result. For instance, Singh and Sharma [31] applied a hybrid approach combining AHP, DEMATEL, and TOPSIS to evaluate the performance of manufacturing organizations. Alves et al. [32] provided a comprehensive approach for reducing uncertainties in decision-making tasks by utilizing multiple multicriteria methods and scenarios in power generation planning. Lin et al. [33] tackled fuzziness and uncertainties in measured data for risk factors with triangular fuzzy sets. Both expert judgments and measured data were considered in the comprehensive weight determination of risk factors. Susmaga et al. [34] demonstrated that TOPSIS and similar distance-based aggregation methods could be successfully illustrated in a plane and interpreted even when the criteria are weighted, regardless of their number. Alinezhad et al. [24] proposed a hybrid multicriteria decision-making (MCDM) model based on triangular fuzzy numbers (TFNs) that combines both subjective and objective approaches. Although hybrid methods aim to improve robustness or incorporate the different facets of a decision problem, engineering managers may still face challenges in their application to UAV swarm performance. These include the potential for increased model complexity, the difficulty of dynamically updating aggregated outputs, and ensuring that the underlying logic remains transparent enough to support accountable decision-making, especially when fusing diverse expert opinions with operational data from the swarm.

Machine learning has also been utilized for aggregating MCDM results. Guo et al. [11] established an NN-MCDM model that could capture the implicit high-order interactions among criteria and their complex nonlinear transformations. Ma and Li [35] leveraged machine learning methods to aggregate numerous MCDM results. Wang et al. [10] proposed a methodology that combines weighting, aggregating, and grouping into a one-stop process. A significant challenge with some advanced computational approaches, however, is that their ’black-box’ nature can hinder adoption by engineering managers who require transparency. This lack of interpretability can be a significant barrier when accountability and clear justification for decisions are paramount. Consequently, there is a pressing need within engineering management for methodologies that not only combine expert experience with empirical data effectively within a single, coherent pipeline, but also maintain a high degree of transparency and interpretability to support managerial oversight and trust.

Table 1 provides a comparative analysis of our proposed framework against the recent MCDM literature across several key dimensions. The comparison considers the source of criterion weights, distinguishing between Objective Weighting Methods (e.g., based on data entropy) and Subjective Weighting Methods (e.g., AHP). Applicability is assessed for both Single Decision-Maker contexts and Group Decision-Making scenarios. Furthermore, we evaluate whether a method constitutes a Unified Pipeline and its degree of Interpretability and Transparency. This final criterion assesses whether a method’s inner workings are clearly explained and theoretically justified, as opposed to functioning as an opaque “black box”. The comparison highlights a key gap in the existing literature: few methods successfully integrate both subjective expert judgment and objective data within a single, transparent, and unified pipeline. Our proposed framework is specifically designed to address this gap by satisfying all of these criteria.

The review of UAV swarm technologies (Section 2.1) highlights the complexity and multifaceted nature of swarm performance, while the assessment of existing MCDM methods (Section 2.2) reveals limitations in their adaptability, integration of heterogeneous information, and transparency for managerial oversight, particularly in dynamic operational contexts. Consequently, there is a clear need, from an engineering management perspective, for a novel decision-making framework that can (1) holistically model the interdependent criteria of UAV swarm performance, (2) systematically integrate initial expert intuition with evolving real-time data, (3) provide interpretable and robust sensitivity analyses to identify critical performance drivers, and (4) support adaptive decision making throughout the lifecycle of UAV swarm operations. Our proposed Bayesian Network-based framework directly addresses this multifaceted gap, offering a unified and adaptable methodology tailored to the unique challenges of evaluating and managing UAV swarm performance.

## 3. Preliminaries

To construct the model, some related theories, including AHP, SOBOL, BN parameter and inference, must be given.

### 3.1. Analytical Hierarchy Process

The AHP estimates priorities from relative deviations in pairwise comparison matrices Saaty [43]. Pairwise comparisons are made using a scale from 1 (equal importance) to 9 (extreme importance). In a tree-like structure, the AHP hierarchy is separated into three levels: goal level, criteria level, and factor level [44]. Assume that the goal level is composed of a goal *G*, the criteria level is composed of *n* criteria of $C=\{C_{1},C_{2},\ldots ,C_{n}\}$, and the factor level is composed of *m* factors $F=\{F_{1},F_{2},\ldots ,F_{m}\}$. The basic steps of AHP are as follows.

(1) The pairwise comparison matrix of *G* is established by(1)$$\begin{array}{c} A^{G}={\left[\alpha_{ij}\right]}_{n\times n}=\left[\begin{array}{cccc} 1 & \alpha_{12} & \ldots & \alpha_{1n}\\ \alpha_{21} & 1 & \vdots & \alpha_{2n}\\ \vdots & \vdots & \ddots & \vdots \\ \alpha_{n1} & \alpha_{n2} & \ldots & 1 \end{array}\right], \end{array}$$
where $A^{G}={\left[\alpha_{ij}\right]}_{n\times n}$ is the matrix of *n* criteria for *G*, and $\alpha_{ij}$ is an indicator for the importance of $C_{i}$ relative to $C_{j}$. Diagonal elements are 1 as they correspond to the importance of each criterion relative to itself, and reciprocal values exist for *i* to *j* and *j* to *i*. Therefore, $\alpha_{ij}=1/\alpha_{ji}$.

(2) The weights of *G* $\Omega^{G-C}=\left(\omega_{G-C_{1}},\ldots ,\omega_{G-C_{n}}\right)$ are computed, where $\omega_{G-C_{i}}$ represents the weight of $C_{i}$.

(3) The consistency of the pairwise comparison matrix is checked.

Considering expert intuition, the judgments in $A^{G}$ may conflict with each other. Thus, we compute consistency ratio $C_{R}$ to assure the validity of $A^{G}$ using (Equation 2)(2)$$\begin{array}{cc} C_{I} & =\frac{\lambda_{\max}-n}{n-1},\\ C_{R} & =\frac{C_{I}}{\gamma }, \end{array}$$
where $\lambda_{\max}$ is the maximal eigenvalue of $A^{G}$, $C_{I}$ is the Consistency Index, and γ is a Random Index value from the dimension of $A^{G}$. The values of γ are given in Table 2 Saaty [43]. The pairwise comparison matrix is acceptable if $C_{R}<0.1$. Otherwise, we need to repeat step 2 until it is acceptable.

Following the processes outlined above, the weights of the remaining factors are calculated.

### 3.2. SOBOL

SOBOL is a variance-based sensitivity analysis method that assesses the contribution of each input to the output. We define the variable symbols and their meanings before describing the method. The uppercase letters *X* and *Y* represent the factors, and the lowercase letters *x* and *y* denote the corresponding values of the factors. For example, $x_{i}$ corresponds to the value of $X_{i}$. In general, *X* is specified as the input variable and *Y* as the output variable. Then, the system *f* can be expressed as (Equation 3)(3)$$\begin{array}{c} Y=f\left(X\right), \end{array}$$
where $X=\{x_{1},x_{2},\ldots ,x_{n}\}$ and *n* is the dimension of the input.

Then, we can obtain the decomposition of system *f* Sun et al. [45]:(4)$$\begin{array}{c} Y=f_{0}+\sum_{i=1}f_{i}+\sum_{i}\sum_{j>i}f_{ij}+\ldots +f_{1,2,\ldots ,n}, \end{array}$$
where $f_{i}=f_{i}\left(X_{i}\right)$, $f_{ij}=f_{ij}\left(X_{i},X_{j}\right)$, $f_{1,2,\ldots ,n}=f_{1,2,\ldots ,n}\left(X_{1},X_{2},\ldots ,X_{n}\right)$ and all the *f* terms in (Equation 4) are conditional independence terms.

The following derivation is based on the classical variance decomposition formula provided in Equation (Equation 5) from Saltelli [46](5)$$\begin{array}{c} V\left(Y\right)=E_{X_{i}}\left(V_{X_{∼i}}\left(Y|X_{i}\right)\right)+V_{X_{i}}\left(E_{X_{∼i}}\left(Y|X_{i}\right)\right), \end{array}$$
where *V* represents the variance operator, and *E* represents the expectation operator. $X_{∼i}$ is the set $\left\{x_{j}|j\neq i\right\}$. (Equation 5) indicates that the larger $V_{X_{i}}\left(E_{X_{∼i}}\left(Y|X_{i}\right)\right)$ is, the more significant the influence of $X_{i}$ on *Y*. Thus, we can decompose the variance of the model output *Y* as follows [46]:(6)$$\begin{array}{c} V\left(Y\right)=\sum_{i}V_{i}+\sum_{i}\sum_{j>i}V_{ij}+\ldots +V_{1,2,\ldots ,n}, \end{array}$$
where $V_{i}=V\left(E\left(Y|X_{i}\right)\right)$, $V_{ij}=V\left(E\left(Y|X_{i},X_{j}\right)\right)-V_{i}-V_{j},\ldots$, $V_{1,2,\ldots ,n}=V(E(Y|X_{1},X_{2},\ldots ,X_{n}))-\sum_{i_{1}}V_{i_{1}}-\sum_{i_{1}}\sum_{i_{2}>i_{1}}V_{i_{1}i_{2}}-\cdots -\sum_{i_{1}}\cdots \sum_{i_{n-1}>\cdots >i_{2}>i_{1}}V_{i_{1}i_{2}\ldots i_{n-1}}$. Then, the sensitivity values of the relevant input variables were calculated by normalizing $V_{i}$ and $V_{ij}$ with the variance of *Y*, and the SOBOL sensitivity indices of the input were obtained by (Equation 7) and (Equation 8) [46](7)$$\begin{array}{cc} s_{i} & =\frac{V_{i}}{V(Y)},i\in \left[1,n\right], \end{array}$$(8)$$\begin{array}{cc} s_{ij} & =\frac{V_{ij}}{V(Y)},i,j\in \left[1,n\right], \end{array}$$
where $s_{i}$ represents the main SOBOL sensitivity indices of $X_{i}$, and $s_{ij}$ represents the second-order SOBOL sensitivity indices $X_{i}$ and $X_{j}$. The magnitude of $s_{i}$ measures the contribution of the input change to the output change. Additionally, the main SOBOL sensitivity indices are considered in this article. Finally, we obtain (Equation 9) by extending (Equation 7) [46].(9)$$\begin{array}{c} s_{i}=\frac{V_{X_{i}}\left(E_{X_{∼i}}\left(Y|X_{i}\right)\right)}{V\left(Y\right)} \end{array}$$

Similar to AHP, the results of SOBOL are referred to as $S^{Y-X}=\left(s_{Y-X_{1}},s_{Y-X_{2}},\ldots ,s_{Y-X_{n}}\right)$.

### 3.3. Bayesian Network

BN is a probabilistic directed acyclic graphical model composed of the structure and parameters. The structure consists of *n* nodes $X=\{X_{1},X_{2},\ldots ,X_{n}\}$, edges, and the parameters. Nodes represent variables, edges represent direct probabilistic dependencies, and parameters measure the strengths of the dependencies. Generally, structure construction, parameter construction, and inference are the three steps of using BN.

First, knowledge representation and structure learning are two basic strategies used in structure construction [12,47]. The former is adopted in this paper, where experts acquire knowledge regarding node relationships.

Once the structure is defined, the parameters, in the form of CPTs for each node, are estimated. This can be based on expert knowledge, learned from data, or a combination of both Yang et al. [48], Guo et al. [49]. For simplicity, we use $\theta_{ijk}$ to represent the parameter, as shown in (Equation 10) [12]:(10)$$\begin{array}{c} \theta_{ijk}=P\left(X_{i}=k|\pi \left(X_{i}\right)=j\right), \end{array}$$
where $k\in \{1,2,\ldots ,r_{i}\}$ is the state of node $X_{i}$, and $j\in \{1,2,\ldots ,q_{i}\}$ is the configuration of the parent set $\pi \left(X_{i}\right)$. In this study, we use maximum a posteriori estimation (MAP) to learn the parameters, which introduces Dirichlet priors and represents expert experience through its hyperparameter $D=\alpha_{ijk}$. Thus, when the posterior distribution of the parameter is maximized, we obtain the estimated value of the parameter as in (Equation 11) [12].(11)$$\begin{array}{c} \theta_{ijk}=\frac{N_{ijk}+\alpha_{ijk}}{\sum_{k=1}^{r_{i}}N_{ijk}+\alpha_{ijk}}, \end{array}$$
where $N_{ijk}$ is the count for records when $X_{i}=k$ and $\pi (X_{i})=j$. The Dirichlet prior hyperparameters can be regarded as the number of virtual samples of parameters.

After acquiring the structure and parameters of the Bayesian Network (BN), inference algorithms can be employed to address various problems [50,51]. BN inference involves computing the posterior probability of a variable given that other variables have been assigned specific values. In this work, we use the variable elimination (VE) algorithm as implemented by Ankan and Textor [52]. VE conducts exact inference by iteratively multiplying and marginalizing factors, starting from the network’s conditional probability tables, according to a fixed elimination order. Once a Bayesian network’s parameters are fully specified, any probability of interest can be computed [53].

## 4. Model Construction

This section details the construction of our Bayesian Network Decision-Making (BNDM) framework, as shown in Figure 3. The framework is built upon a probabilistic graphical structure designed to explicitly model a multicriteria decision problem, leveraging the natural correspondence between the hierarchical decomposition used in MCDM methods like AHP and the architecture of a Bayesian Network. By representing factors, criteria, and the overall goal as nodes in a multi-level graph, this structure allows for quantitative reasoning about the propagation of influence from low-level performance metrics to high-level strategic objectives.

The following subsections first formally define this hierarchical network structure, then introduce the two core analytical methods that operate upon it, and finally integrate everything into a complete, step-by-step procedural framework.

### 4.1. The Multi-Level Hierarchical BN Structure

To systematically translate a multicriteria decision-making problem into a probabilistic inference model, we first formally define the multi-level hierarchical Bayesian Network structure that serves as the backbone of our framework.

This architecture explicitly organizes the evaluation system into three distinct levels, as shown in Figure 10. At the apex is the Goal Layer, consisting of a single leaf node, *G*, which represents the final evaluation objective (e.g., ‘Swarm Performance’). This goal is directly influenced by the Criteria Layer, a set of intermediate nodes, $C=\{C_{1},\ldots ,C_{n}\}$, where each $C_{i}$ represents a high-level evaluation criterion like ‘Swarm Mobility.’ Finally, the foundation of the network is the Factor Layer, which contains all root nodes, $F=\{F_{1},\ldots ,F_{m}\}$. Each factor, such as ‘Path Planning Efficiency’ represents a fundamental performance metric and serves as a parent to one and only one criterion node.

Based on this hierarchical structure, the joint probability distribution of the entire network, P(G,C,F), can be precisely factorized into a product of local conditional probabilities according to the chain rule of BN. This factorization is the mathematical core of the entire framework and its final form is expressed as(12)$$P\left(G,C,F\right)=P\left(G|C_{1},\ldots ,C_{n}\right)\cdot \prod_{i=1}^{n}P\left(C_{i}|\pi \left(C_{i}\right)\right)\cdot \prod_{j=1}^{m}P\left(F_{j}\right)$$
where P(G,C,F) represents the joint probability of all nodes in the network, $P(G|C_{1},\ldots ,C_{n})$ is the conditional probability for the Goal Layer, quantifying how the criteria jointly influence the final goal, and $\prod_{i=1}^{n}P\left(C_{i}|\pi \left(C_{i}\right)\right)$ is the product of conditional probabilities for the Criteria Layer. Here, $\pi (C_{i})$ denotes the set of parent factor nodes for the criterion $C_{i}$ and $\prod_{j=1}^{m}P\left(F_{j}\right)$ is the product of prior probabilities for the Factor Layer (leaf nodes).

The significance of this factorization is that it decomposes a complex, high-dimensional joint probability problem into a combination of smaller, more manageable local probability tables. This decomposability, combined with the conditional independence assumptions inherent in the model, justifies simplifying our subsequent analysis to focus on generic parent–child sub-networks. These sub-networks can represent either the ‘Factor-to-Criterion’ relationships ($F\to C_{i}$) or the ‘Criteria-to-Goal’ relationship (C→G). Therefore, for the sake of generality and notational simplicity in the subsequent mathematical derivations, we will use the generic symbols $X=\{X_{1},\ldots ,X_{k}\}$ to represent a set of parent nodes in an arbitrary sub-network, and *Y* to represent its corresponding child node. This approach ensures that our derived methods (such as Hierarchical Bayesian Network Sensitivity (HBNS) and Multicriteria Hierarchical Bayesian Network Sensitivity (M-HBNS)) are generalizable and can be applied to any hierarchical relationship within the network. Moreover, the entire BNDM framework is, in essence, built around learning and constructing these local probability terms from expert knowledge and data.

### 4.2. Hierarchical Bayesian Network Sensitivity Method

Once the hierarchical network structure is defined, the primary analytical task is to quantify the influence of each parent node $X_{i}$ on its child node *Y*. To this end, the Hierarchical Bayesian Network Sensitivity (HBNS) method is developed. It provides an analytical solution for computing sensitivity indices by adapting principles from variance decomposition. This approach enables the exact computation of sensitivities directly from the model’s parameters, thereby obviating the need for computationally expensive Monte Carlo simulations.

To demonstrate the derivation process in a hierarchical network, we define a simple BN with *n* parents $\left\{X_{1},X_{2},\ldots ,X_{n}\right\}$ of node *Y*. For simplicity in expressing conditional probabilities, $P\left(x_{i}=j\right)=P\left(x_{i}^{j}\right)$ is specified.

First, we calculate the mean value of *Y* and $Y^{2}$ as follows:(13)$$\begin{array}{cc} E\left(Y\right) & =\sum_{k=1}^{r}\sum_{j=1}^{\prod_{i=1}^{n}q_{i}}y^{k}\cdot P\left(y^{k}|x_{1}^{x_{j1}},\ldots ,x_{n}^{x_{jn}}\right) \end{array}$$(14)$$\begin{array}{cc} E\left(Y^{2}\right) & =\sum_{k=0}^{r-1}\sum_{j=1}^{\prod_{i=1}^{n}q_{i}}{\left(y^{k}\right)}^{2}\cdot P\left(y^{k}|x_{1}^{x_{j1}},\ldots ,x_{n}^{x_{jn}}\right) \end{array}$$

Similar to (Equation 10), where $k\in \left\{0,1,\ldots ,r-1\right\}$ is the state of *Y*, $j\in \{1,2,\ldots ,\prod_{i=1}^{n}q_{i}\}$ is the configuration of the parent set $\left\{X_{1},X_{2},\ldots ,X_{n}\right\}$, and $x_{i}\in \left\{1,2,\ldots ,q_{i}\right\}$. Further, we can obtain the variance of *Y* using (Equation 13) and (Equation 14).(15)$$\begin{array}{c} V\left(Y\right)=E\left(Y^{2}\right)-E^{2}\left(Y\right) \end{array}$$

It is complex to denote V(Y) with (Equation 13) and (Equation 14). So, $P(y^{k})$ can be obtained by BN inference to simplify (Equation 15)(16)$$V\left(Y\right)=\sum_{k=0}^{r-1}k^{2}\left(P\left(y^{k}\right)-P^{2}\left(y^{k}\right)\right)$$

Equations (Equation 17)–(Equation 19) are necessary for the further derivation of (Equation 9)(17)$$\begin{array}{c} V_{X_{i}}\left(E_{X_{∼i}}\left(Y|X_{i}\right)\right)=E_{X_{i}}\left({\left(E_{X_{∼i}}\left(Y|X_{i}\right)\right)}^{2}\right)-{E_{X_{i}}}^{2}\left(E_{X_{∼i}}\left(Y|X_{i}\right)\right), \end{array}$$(18)$$\begin{array}{c} E_{X_{i}}\left(E_{X_{∼i}}\left(Y|X_{i}\right)\right)=\sum_{j=0}^{q_{i}-1}P(x_{i}^{j})\cdot E_{X_{∼i}}\left(y|x_{i}^{j}\right), \end{array}$$(19)$$\begin{array}{c} E_{X_{∼i}}\left(Y|X_{i}\right)=\sum_{k=0}^{r}\sum_{t=1}^{\prod_{m=1,m\neq i}^{n}q_{i}}p(X_{∼i}=j_{t})\cdot k\cdot p\left(y^{k}|x_{i}^{j_{i}},X_{∼t}=j_{t}\right), \end{array}$$
where $j_{i}$ is the *j*-th value of $X_{i}$ and $j_{t}$ is the *t*-th value of the node combination except $X_{i}$.

The simplification of (Equation 19), which is necessary for deriving an analytical expression for sensitivity, relies on established principles for decomposing joint conditional probabilities in BNs under certain assumptions, as formalized in Theorem 1.

**Theorem 1.** *Zhang and Poole [54] Let Y be a node in a BN and let $X_{1}$, $X_{2}$, …,$X_{n}$ be the parents of Y. If $X_{1}$, $X_{2}$, …,$X_{n}$ are causally independent (d-separation [12]) and Y has m states $y_{1},y_{2},\ldots ,y_{m}$, the BN parameters $P(Y|X_{1},X_{2},\ldots ,X_{n})$ can be obtained from the conditional probability $P(Y|X_{1})$, $P(Y|X_{2})$,…, $P(Y|X_{n})$ through* (Equation 20).
(20)$$\begin{array}{c} P\left(y^{y_{j}}|x_{1},\ldots ,x_{n}\right)=\sum_{y_{j}=y_{1}*y_{2}*\ldots *y_{k}}P\left(y^{y_{1}}|x_{1}\right)P\left(y^{y_{2}}|x_{2}\right)\ldots P\left(y^{y_{m}}|x_{n}\right) \end{array}$$
*Here, * is the combination operator of Y, and j=1,2,3,…,m.*


Theorem 1 presents a general formulation of BN parameters and conditional probability, which decompose $P(Y|X_{1},X_{2},\ldots ,X_{n})$ in P(Y∣X). Moreover, we give the specific form under the hierarchical BN structure in Corollary 1.

**Corollary 1.** 
*The joint probability can be calculated according to the marginal probability when the structure of BN is hierarchical. Thus, the specific form is as follows:*

(21)
$$\begin{array}{c} \left(y^{y_{j}}|x_{1}^{x_{1i}},x_{2}^{x_{2i}},\ldots ,x_{n}^{x_{ni}}\right)=\frac{P\left(y^{y_{j}}|x_{1}^{x_{1i}}\right)P\left(y^{y_{j}}|x_{2}^{x_{2i}}\right)\cdots P\left(y^{y_{j}}|x_{n}^{x_{ni}}\right)}{P{\left(y^{y_{j}}\right)}^{n-1}} \end{array}$$



**Proof.** First, (Equation 22) holds according to the conditional probability formula.(22)$$\begin{array}{c} P\left(y^{y_{j}}|x_{1}^{x_{1i}},x_{2}^{x_{2i}},\ldots ,x_{n}^{x_{ni}}\right)=\frac{P\left(y^{y_{j}},x_{1}^{x_{1i}},x_{2}^{x_{2i}},\ldots ,x_{n}^{x_{ni}}\right)}{P\left(x_{1}^{x_{1i}},x_{2}^{x_{2i}},\ldots ,x_{n}^{x_{ni}}\right)} \end{array}$$Then, since $X_{1}$, $X_{2}$, $X_{3}$, …, $X_{n}$ are independent of each other, (Equation 23) holds.(23)$$\begin{array}{c} P\left(x_{1}^{x_{1i}},x_{2}^{x_{2i}},\ldots ,x_{n}^{x_{ni}}\right)=P\left(x_{1}^{x_{1i}}\right)P\left(x_{2}^{x_{2i}}\right)\ldots P\left(x_{n}^{x_{ni}}\right) \end{array}$$Further, $\left(X_{1},Y\right)$, $\left(X_{2},Y\right)$, $\left(X_{3},Y\right)$, …, $\left(X_{n},Y\right)$ are independent of each other, and thus, (Equation 24) holds.(24)$$\begin{array}{c} P\left(y^{y_{j}},x_{1}^{x_{1i}},x_{2}^{x_{2i}},\ldots ,x_{n}^{x_{ni}}\right)=\frac{P\left(y^{y_{j}},x_{1}^{x_{1i}}\right)P\left(y^{y_{j}},x_{2}^{x_{2i}}\right)\cdots P\left(y^{y_{j}},x_{n}^{x_{ni}}\right)}{P{\left(y^{y_{j}}\right)}^{n-1}} \end{array}$$Finally, (Equation 21) is obtained using (Equation 23) and (Equation 24) due to P(Y∣X)=P(XY)/P(X). □

According to Corollary 1, we can simplify (Equation 19) as (Equation 25)(25)$$\begin{array}{c} E_{X_{∼i}}\left(Y|X_{i}\right)=\sum_{k=0}^{r-1}\sum_{t=1}^{\prod_{m=1,m\neq i}^{n}q_{i}}P(X_{∼i}=j_{t})\cdot k\cdot P\left(y^{k}|X_{∼i}=j_{t}\right)P\left(y^{k}|x_{i}^{j_{i}}\right)/P\left(y^{k}\right). \end{array}$$

Then, (Equation 26) holds according to the law of total probability.(26)$$\begin{array}{c} E_{X_{∼i}}\left(Y|X_{i}=x_{i}\right)=\sum_{k=0}^{r-1}k\cdot P(Y=k|X_{i}=x_{i}), \end{array}$$

Simultaneously with (Equation 17), (Equation 18), and (Equation 26), we can obtain (Equation 27).(27)$$\begin{array}{c} V_{X_{i}}\left(E_{X_{∼i}}\left(Y|X_{i}\right)\right)\\ =\sum_{j=0}^{q_{i}-1}P\left(X_{i}=j\right)\cdot [\sum_{k=0}^{r-1}jP\left(Y=k|X_{i}=x_{i}\right)-\sum_{j=0}^{q_{i}-1}P(X_{i}=x_{i})kP\left(Y=k|X_{i}=j\right){]}^{2} \end{array}$$

Furthermore, considering (Equation 9), (Equation 16), and (Equation 27), (Equation 28) is given.(28)$$\begin{array}{c} s_{i}=\frac{\sum_{j=0}^{q_{i}-1}P(X_{i}=j)}{\sum_{k=0}^{r-1}k^{2}\left(P\left(y^{j}\right)-P^{2}\left(y^{j}\right)\right)}\cdot [\sum_{k=0}^{r-1}jP\left(Y=k|X_{i}=x_{i}\right)-\sum_{j=0}^{q_{i}-1}P(X_{i}=x_{i})kP\left(Y=k|X_{i}=j\right){]}^{2} \end{array}$$

In this paper, we define that *X* and *Y* have two states, which are $\left\{bad,good\right\}$ represented by the numbers {0,1}. Under this assumption, we can further simplify (Equation 28) as (Equation 29).(29)$$\begin{array}{c} s_{i}^{\prime }=\frac{\sum_{j}^{0,1}{\left(P\left(y^{1}|x_{i}^{j}\right)-P\left(y^{1}\right)\right)}^{2}\cdot P\left(x_{i}^{j}\right)}{\left(P\left(y^{1}\right)-P^{2}\left(y^{1}\right)\right)}, \end{array}$$
where $s_{i}^{\prime }$ refers to the main SOBOL indices obtained in this case using conditional probability calculations. Equations (Equation 28) and (Equation 29) exhibited a method for calculating the main SOBOL sensitivity indices.

Building upon these derivations, Theorem 2 formally articulates the HBNS method for calculating sensitivity indices directly from BN parameters under hierarchical network conditions.

In this work, we assume that there is a mapping relationship between the values obtained by the MCDM and SA. This assumption is common sense, and we present it as Definition 1.

**Definition 1.** *The weight measures the relative importance of the factors. AHP applies expert intuition to access $\Omega =(\omega_{1},\omega_{2},...,\omega_{n})$, and HBNS introduces conditional variance to obtain $S^{\prime }=\left({s_{1}}^{\prime },{s_{2}}^{\prime },...,{s_{n}}^{\prime }\right)$. Their mapping relationship is expressed by* (Equation 30)
(30)$$\begin{array}{c} \left(\omega_{1},\omega_{2},\ldots ,\omega_{n}\right)=\frac{\mu }{\min(scope_{i})}\left({s_{1}}^{\prime },{s_{2}}^{\prime },\ldots ,{s_{n}}^{\prime }\right), \end{array}$$*where $scope_{i}\in \left[0,1\right]$ is the scope of $s_{i}^{\prime }$, μ is the scaling factor, and the default value is 1.*

**Theorem 2.** *Consider a Bayesian Network G=(L(N,E),θ), where L denotes the set of nodes N and edges E in the network, and θ denotes the parameters of the network, specifically the conditional probability tables. Suppose there exists a system f that can be expressed *(Equation 3). *X and Y can be represented by the nodes N. We define an operator H that measures the contribution of X to Y. Then, we have the following expression:*(31)$$S^{Y-X}=H\left(\left(L\left(N,E\right),\theta \right)\right)$$*Under the assumption of parent node independence (as utilized in Corollary 1), the main sensitivity indices $s_{i}$ can be derived using* (Equation 28). *This expression further simplifies to* (Equation 29) *when the nodes $X_{i}$ and Y are binary.*

This efficiency is a key consideration for engineering managers, as it ensures that the framework remains scalable and practical for evaluating increasingly complex swarm configurations without prohibitive computational overhead. For a naïve Bayesian network with *N* binary nodes, if no independence assumptions are made, the number of parameters required for the CPTs grows exponentially, i.e., $O(2^{N})$. However, by introducing the assumption of parent node independence, each node’s CPT can be decomposed into the product of its marginal conditional probabilities, significantly reducing the number of parameters to O(N). This reduction in the parameter space not only greatly enhances the scalability of our model for multicriteria decision making in UAV swarm management, but also dramatically improves computational efficiency.

### 4.3. Multicriteria Hierarchical Bayesian Network Sensitivity Analysis

While HBNS establishes a method to derive sensitivity indices from BN probabilities, a crucial step for practical application, particularly when initializing a model with expert knowledge, is the inverse process: deriving BN conditional probabilities from expert-assigned importance weights. This subsection introduces the Multicriteria Hierarchical Bayesian Network Sensitivity (M-HBNS) method, which is designed to systematically convert such weights into consistent BN CPT parameters by leveraging the variance decomposition principles.

First, to obtain the conditional probabilities with respect to $s_{i}^{\prime }$, we use (Equation 29) and constraints to build a system of equations, as shown in (Equation 32).(32)$$\left\{\begin{array}{cc} \; & \frac{\sum_{j}^{0,1}{\left(P\left(y^{1}|x_{i}^{j}\right)-P\left(y^{1}\right)\right)}^{2}\times P\left(x_{i}^{j}\right)}{\left(P\left(y^{1}\right)-P^{2}\left(y^{1}\right)\right)}=s_{i}^{\prime }\\ \; & \sum_{j}^{0,1}P\left(x_{i}^{j}\right)=1\\ \; & \sum_{j}^{0,1}P\left(y^{k}|x_{i}^{j}\right)P\left(x_{i}^{j}\right)=P\left(y^{k}\right),k=0,1 \end{array}\right.$$

Then, the results of (Equation 33) are as follows:(33)$$\begin{array}{c} \left\{\begin{array}{c} P\left(y^{1}|x_{i}^{0}\right)=\frac{P\left(y^{1}\right)+\left(P\left(y^{1}\right)-1\right)\left(P\left(y^{1}\right)-H\right)}{P\left(x_{i}^{0}\right)}\\ P\left(y^{1}|x_{i}^{1}\right)=P\left(y^{1}\right)-H \end{array}\right.,P\left(y^{1}|x_{i}^{0}\right)>P\left(y^{1}|x_{i}^{1}\right)\\ \left\{\begin{array}{c} P\left(y^{1}|x_{i}^{0}\right)=\frac{P\left(y^{1}\right)+\left(P\left(y^{1}\right)-1\right)\left(P\left(y^{1}\right)+H\right)}{P\left(x_{i}^{0}\right)}\\ P\left(y^{1}|x_{i}^{1}\right)=P\left(y^{1}\right)+H \end{array}\right.,P\left(y^{1}|x_{i}^{0}\right)<P\left(y^{1}|x_{i}^{1}\right) \end{array},$$
where $H=\sqrt{s_{i}^{\prime }P\left(x_{i}^{0}\right)P\left(y^{1}\right)\left(P\left(x_{i}^{0}\right)-1\right)\left(P\left(y^{1}\right)-1\right)}/P\left(x_{i}^{0}\right)$ and $P\left(y^{1}|x_{i}^{0}\right)<P\left(y^{1}|x_{i}^{1}\right)$ imply that the effect of $X_{i}$ on *Y* is positive and $X_{i}$ and *Y* are positively correlated. In contrast, $X_{i}$ and *Y* are negatively correlated. We set $o_{i}=1$ to indicate a positive correlation and $o_{i}=0$ to indicate a negative correlation between $X_{i}$ and *Y*. According to the results of (Equation 32), $P(Y|X_{i})$ is calculated when $P(X_{i})$, P(Y), and $s_{i}^{\prime }$ are known.

The mathematical formulation for this inverse mapping from sensitivity indices (weights) to conditional probabilities is encapsulated in Theorem 3.

**Theorem 3.** *Consider the Bayesian Network G=(L(N,E),θ) and the system f expressed by* (Equation 3). *Let $\Omega =\{\omega_{1},\omega_{2},\ldots ,\omega_{n}\}$ denote the set of weights, and U={P(n)∣n∈N} represent the set of probabilities associated with the nodes. We define an operator R that maps the relationship between the CPTs θ and the weights* Ω. *Then, we have the following expression:*
(34)θ=R(L(N,E),Ω,U)*When the parent nodes are independent and the size is equal to 2, the parameters can be derived by* (Equation 33).

Additionally, we need to constrain the range of variables in (Equation 33):

The probability is a real number between 0 and 1, and the square root must be greater than zero, which is projected as(35)$$\left\{\begin{array}{c} 1<P\left(y^{1}|x_{i}^{0}\right)<0\\ 1<P\left(y^{1}|x_{i}^{1}\right)<0\\ \Delta =P\left(x_{i}^{0}\right)P\left(y^{1}\right)s_{i}^{\prime }\left(P\left(x_{i}^{0}\right)-1\right)\left(P\left(y^{1}\right)-1\right)\geq 0 \end{array}\right.$$

The solution space of $P(x_{i}^{1})$, $P(y^{1})$, and $s_{i}^{\prime }$ is shown in Figure 4.

In Figure 4, the solution space is the solid bounded by the surface and the $P\left(x_{i}^{1}\right)-P\left(y^{1}\right)$ plane. When the value of $s_{i}^{\prime }$ is larger, the value range of $P(x_{i}^{1})$ and $P(y^{1})$ is smaller, and vice versa. In the case where $s_{i}^{\prime }$ is known, the value range of $P(x_{i}^{1})$ is the largest when $P(y^{1})=0.5$, and when $P(y^{1})$ is closer to 0 and 1, its value range decreases and distributes on both sides of 0.5. $P(y^{1})$ has the same features, when $P(x_{i}^{1})=0.5$. In the plane composed of $x_{i}^{1}-s_{i}^{\prime }$, different curves reflect the relationship between $x_{i}^{1}$ and $s_{i}^{\prime }$ when $y^{1}$ takes different values. When $y^{1}$ is relatively large, $x_{i}^{1}$ tends to be smaller as $s_{i}^{\prime }$ becomes larger. In the opposite case, $x_{i}^{1}$ tends to be larger as $s_{i}^{\prime }$ becomes larger.

This relationship, depicted in Figure 4, aligns with an intuitive understanding: a factor ($X_{i}$) is considered highly influential if its state significantly alters the probability of the desired outcome (*Y*). Engineering managers can leverage this understanding when providing initial expert judgments (hyperparameters for $P(x_{i}^{1})$ and $s_{i}^{\prime }$), ensuring that the model’s initial configuration reflects realistic operational sensitivities.

### 4.4. BNDM Framework

The preceding subsections defined the model’s structure and its core analytical components. We now integrate them into a comprehensive, four-phase procedure for practical application, as illustrated in Figure 5. We used a two-round process: in Round 1, experts proposed/edited the criteria–factor lists adopting the literature that we cite before; in Round 2, disagreements were resolved in a moderated session by majority with the rationale recorded. This framework guides a manager from initial problem structuring with expert knowledge to a refined, data-driven performance model.

The four-phase procedure is outlined as follows.

Phase 1 Expert evaluation: This phase leverages the structured AHP methodology to systematically capture initial expert judgments on criteria importance, forming the foundational expert-driven input for the subsequent BN construction. This ensures that managerial priorities are embedded from the outset.

**Step 1:** After discussion, experts determine the criteria $C=\{C_{1},C_{2},C_{3},\ldots ,C_{n}\}$ and factors $F=\{F_{1},F_{2},F_{3},\ldots ,F_{m}\}$ that should be considered in different goals *G*.

**Step 2:** Organize the identified goal, criteria, and factors into a hierarchical structure G−C−F. This AHP-based hierarchy provides a clear visual representation of performance dependencies and forms the basis for structured expert judgment elicitation, aligning the model with how managers often conceptualize complex problems.

**Step 3:** The criteria and factors are used to construct the pairwise comparison matrix $A^{G-C}$, $A^{C_{1}-F}$, $A^{C_{2}-F}$,…, $A^{C_{n}-F}$.

**Step 4:** Compute weight $\Omega^{G-C}$, $\Omega^{C_{1}-F}$, $\Omega^{C_{2}-F}$, …, $\Omega^{C_{n}-F}$.

**Step 5:** Calculate the consistency index and go to the next step if the requirements are met.

Phase 2 M-HBNS: This phase translates the qualitative hierarchy from AHP into a quantitative, probabilistic model.

**Step 6:** Translate the AHP hierarchy G−C−F directly into a multi-level hierarchical BN Structure. Each element (goal, criterion, factor) from the AHP becomes a node in the BN, and the hierarchical relationships define the directed edges in the DAG. This step formally encodes the multicriteria evaluation problem into a probabilistic model.

**Step 7:** Using the M-HBNS method, the AHP weights Ω are converted into the initial CPTs that parameterize the BN, effectively translating expert beliefs into a probabilistic format.

**Step 8:** Calculate the scope and value of $s_{i}^{\prime }$. Here, ${\mathrm{scope}}_{i}$ denotes the admissible interval of the target sensitivity $s_{i}^{\prime }$ that is compatible with the probability constraints and the chosen correlation sign $o_{i}$. In practice, ${\mathrm{scope}}_{i}$ is obtained from the feasibility conditions in Equation (Equation 35), applied to Equation (Equation 33): it is the set of $s_{i}^{\prime }$ values for which there exist conditional probabilities $P\left(y|x_{i}\right)\in \left[0,1\right]$ and marginals $P\left(x_{i}\right),P\left(y\right)$ satisfying the BN constraints. The scalar μ then rescales sensitivities onto a common decision scale; throughout, we set μ=1 and normalize by $\min({\mathrm{scope}}_{i})$ for a conservative scaling.

**Step 9:** Employ the M-HBNS method (Equations (Equation 33) and (Equation 35)) to calculate the initial conditional probabilities $P(Y|X_{i})$ for each parent–child relationship in the BN, based on $P(X_{i})$, P(Y), $o_{i}$, and the target $s_{i}^{\prime }$. This critical step translates abstract importance weights into concrete probabilistic parameters for the BN.

**Step 10:** Utilize the derived $P(Y|X_{i})$ values and Corollary 1 to construct the complete CPTs for all nodes in the BN. This completes the initialization of the BN with expert-derived parameters, creating a baseline model that reflects current managerial understanding.

Thus far, the structure and parameters of BN are constructed in steps 6 and 10, respectively.

Phase 3 Parameter construction: This phase focuses on refining the expert-initialized BN by incorporating empirical evidence from UAV swarm operations, allowing the model to adapt and improve its accuracy over time.

**Step 11:** Convert the initial CPTs (from Step 10) into Dirichlet hyperparameters $\alpha_{ijk}$ for Equation (Equation 11). As new operational data ($N_{ijk}$ counts) from UAV swarm missions becomes available, MAP estimation is employed to update the BN CPTs. This systematically integrates empirical evidence with initial expert beliefs, allowing managers to evolve the model from a primarily expert-driven one to a more data-informed one.

Phase 4 Refined Sensitivity Analysis using HBNS: The final phase uses the data-updated BN to re-evaluate factor sensitivities, providing managers with refined insights into current performance drivers.

**Step 12:** After learning the BN parameters, $P(Y|X_{i})$, $P(X_{i})$ and P(Y) change and their new values are obtained by VE. Then, (Equation 29) is used to calculate the optimized results $s_{i}^{opt}$. These updated indices reflect the influence of factors based on the latest available evidence.

**Step 13:**$s_{i}^{opt}$ is normalized to the final results of BNDM $S^{opt}=\left(s_{1}^{opt},s_{2}^{opt},\ldots ,s_{n}^{opt}\right)$. These refined indices provide engineering managers with an up-to-date understanding of the most critical drivers of UAV swarm performance, guiding targeted interventions, resource adjustments, or strategic reviews.

The flowchart of the BNDM is given in Figure 5. The multi-phase BNDM procedure detailed above outlines a systematic workflow for integrating expert knowledge with empirical data to produce a refined assessment of UAV swarm performance factor sensitivities. Theorem 4 provides a formal aggregation of this entire process, illustrating how initial expert judgments (represented by weights Ω) are transformed into BN parameters, subsequently updated by data *D*, and finally used to derive optimized sensitivity indices $S^{\mathrm{opt}}$. This theorem highlights the BNDM framework’s core capability of achieving a data-informed synthesis.

**Theorem 4.** *Consider the Bayesian Network G=(L(N,E),θ) and the system f expressed by* (Equation 3). *Let $\Omega =\{\omega_{1},\omega_{2},\ldots ,\omega_{n}\}$ denote the set of weights, U={P(n)∣n∈N} represent the set of probabilities associated with the nodes, and $D=\{N_{ijk}\}$ denote the data. We define an operator H that measures the contribution of X to Y considering the data and expert judgments. Then, we have the following expression:*
(36)$$\alpha_{ijk}=R\left(L\left(N,E\right),\Omega ,U\right)$$*Given operational data $D=\{N_{ijk}\}$, where $N_{ijk}$ are observed counts, the posterior CPT parameters ${\hat{\theta }}_{ijk}$ are obtained via MAP estimation* (Equation 11)*:*
(37)$$\begin{array}{c} {\hat{\theta }}_{ijk}=\frac{N_{ijk}+\alpha_{ijk}}{\sum_{k^{\prime }=1}^{r_{i}}\left(N_{ijk^{\prime }}+\alpha_{ijk^{\prime }}\right)} \end{array}$$
*Let $\hat{\theta }$ be the set of these updated CPTs. Finally, the optimized sensitivity indices $S^{\mathrm{opt}}$ are derived from the data-updated BN using the HBNS operator H (Theorem 2):*

(38)
$$S^{opt}=H\left(L\left(N,E\right),\hat{\theta }\right)$$


*Thus, the entire BNDM process can be viewed as an overarching function B:*

(39)
$$S^{opt}=B\left(L\left(N,E\right),\Omega ,U,D\right)$$


*This demonstrates the framework’s capacity to integrate expert intuition (Ω,U) with empirical data (D) within a unified Bayesian pipeline to produce refined sensitivity assessments.*


In Theorem 4, Ω is derived from expert judgments based on MCDM and thus represents expert intuition. *D* is the set of $N_{ijk}$, denoting the data. Based on this formulation (Equation 38), *B* integrates expert intuition and data within a BN, or in other words, within a pipeline.

Note that, when S=Ω, the formulation (Equation 31) in Theorem 2 and the formulation (Equation 34) in Theorem 3 are not strictly inverse functions. However, when *U* is fixed, they can be considered inverse functions, which can be represented as(40)$$R\left(L\left(N,E\right),\Omega ,U\right)=H^{-1}\left(L\left(N,E\right),\theta ,U\right)$$

In summary, this section has detailed the BNDM model, a comprehensive framework that integrates expert knowledge with data-driven learning through its core HBNS and M-HBNS components. By establishing the formal structure, analytical methods, and a procedural workflow, we have laid the foundation for a transparent, adaptable, and robust tool for UAV swarm performance evaluation. The subsequent section will present a comprehensive experimental evaluation designed to validate the framework’s effectiveness and practical applicability.

## 5. Experimental Evaluation

This section presents a three-part experimental evaluation to validate our proposed framework. First, we demonstrate the effectiveness and efficiency of the HBNS method by comparing it against the traditional SOBOL method. Second, we illustrate how the framework’s parameter learning capability allows it to refine results as the volume of available data increases. Finally, we apply the complete BNDM framework to a comprehensive UAV swarm performance evaluation scenario to validate its practical utility in a realistic decision-making context.

### 5.1. Comparison of the HBNS and SOBOL

First, we compare the HBNS and SOBOL sensitivity analysis methods by applying them to a BN with three parent nodes and one child node, as shown in Figure 6. Table 3 lists the parameters of the BN used in this experiment.

The SOBOL method, implemented using the SALib Herman and Usher [55] in Python 3.8.10, follows these steps:(1)Sample the input using Markov chain Monte Carlo (MCMC).(2)Compute the sampled input value of the model.(3)Calculate the sensitivity values one by one.

Given that the BN model does not directly yield exact values through the input, we propose a method for generating discrete binary values using random numbers. Assume a set of results for MCMC sampling inputs as $\left\{x_{i}^{\prime }|x_{i}^{\prime }\in \left(0,1\right),i=1,2,\ldots ,n\right\}$. The remaining steps are as follows:

(1)Obtain $P(x_{i}^{0})$ through BN inference.(2)Generate a random number $x_{i}^{\prime }$. If $x_{i}^{\prime }<P\left(x_{i}^{0}\right)$, set $x_{i}$ to 0; otherwise, set $x_{i}$ to 1, resulting in $x_{i}\in \left\{0,1\right\}$.(3)Use Variable Elimination (VE) to get $P(y^{0}|x_{1},x_{2},\ldots ,x_{n})$.(4)Generate a random number $y_{i}^{\prime }\in \left(0,1\right)$. If $y_{i}^{\prime }<P\left(y^{0}|x_{1},x_{2},\ldots ,x_{n}\right)$, set $y_{i}$ to 0; otherwise, set $y_{i}$ to 1, resulting in $y_{i}\in \left\{0,1\right\}$.

The specific process is shown in Figure 7.

All the data used in this subsection are synthetic and generated from the BN in Figure 4 with the CPTs listed in Table 2. We evaluate four sample sizes $N\in \{2^{10},2^{13},2^{16},2^{19}\}$ and repeat each setting 10 times with a fixed random seed for reproducibility. SOBOL and HBNS are computed on the same synthetic draws to ensure a fair comparison. Figure 8a–d show the results of SOBOL $S^{Y-X}$ and HBNS $S^{\prime Y-X}$ under different data sizes. A box plot was used to display the statistical characteristics of the results, including the minimum, maximum, median, first quartile, third quartile, and outliers. Figure 8 compares the sensitivity indices of $X_{1}$, $X_{2}$, and $X_{3}$ under different sample sizes using SOBOL and HBNS. The Y axis represents the computed sensitivity index, while the X-axis indicates different variables within the Bayesian network. As shown in Figure 8c,d, SOBOL’s results begin to stabilize with fewer outliers when the sample size exceeds $2^{16}$, providing relatively accurate weights for factors $\left\{X_{1},X_{2},X_{3}\right\}$. For a sample size of $2^{19}$, the average SOBOL sensitivity indices are {0.880814,0.105229,0.000103}. In contrast, the HBNS method analytically derived indices of {0.87386269,0.11992151,0.0062158}. The close agreement between these results validates the HBNS approach as an accurate analytical alternative to Monte Carlo-based SOBOL for this BN structure.

The discrete samples for SOBOL are the same synthetic data described above, and the HBNS indices are computed analytically from the same BN, enabling a one-to-one comparison under identical data. These simulation experiments confirmed the correctness of Theorem 2, demonstrating that the weights calculated using this Theorem are consistent with those obtained by SOBOL.

Furthermore, Figure 9 demonstrates a key advantage of HBNS when combined with expert-informed priors and parameter learning: it can achieve robust and centralized sensitivity results even with substantially smaller datasets compared to a purely data-driven SOBOL analysis (cf. Figure 8a). This efficiency is particularly valuable in engineering management contexts, such as the early phases of UAV swarm deployment, where extensive experimental data collection can be prohibitively expensive or time-consuming. The ability to leverage structured expert knowledge to reduce initial data dependency allows for quicker, more cost-effective performance insights and faster iterative learning cycles.

Thus, HBNS requires fewer data to achieve the same precision compared with SOBOL. Additionally, SOBOL has limitations when the model is not generative, requiring tricks to convert the model into a generative one, such as the method proposed here or using neural networks to fit the data. However, HBNS does not have such strict requirements. Introducing expert experience can reduce the need for data and integrate both data and expert intuition into the same evaluation system when certain factors cannot be observed or quantified. This characteristic is particularly beneficial in engineering management contexts where acquiring extensive experimental data for new technologies like UAV swarms can be costly and time-consuming. The ability of HBNS to integrate expert experience effectively reduces this data dependency, allowing for quicker and more resource-efficient initial performance assessments.

### 5.2. UAV Swarm Performance Evaluation

To demonstrate the BNDM framework’s practical utility, we apply it to a comprehensive UAV swarm performance evaluation scenario. This example walks through all four phases of the framework, from eliciting expert knowledge to generating data-driven, actionable insights.

#### 5.2.1. Phase 1: Eliciting Expert Knowledge via AHP

**Steps 1 and 2 (Construct hierarchical structure):** Through consultation with four domain experts and a comprehensive literature review, a hierarchical structure for UAV swarm performance evaluation was established, as illustrated in Figure 10. A core principle guiding this structure, pertinent to effective engineering management, is the recognition that performance assessment must be context-specific. It is crucial to evaluate both the individual UAV capabilities for designated roles and the swarm’s collective coordination efficacy. A monolithic evaluation metric is inadequate for diverse mission profiles and applying a uniform evaluation metric to UAVs assigned to diverse tasks is impractical. Therefore, our assessment framework emphasizes evaluating proficiency against specific assignments and objectives.

**Step 3**: Pairwise comparison matrices $A^{G-C}$, $A^{C_{1}-F}$, $A^{C_{2}-F}$, …, $A^{C_{5}-F}$ are established, as shown in Table 4, Table 5, Table 6, Table 7, Table 8 and Table 9.

**Step 4**: Calculate the weights $\Omega^{G-C}$, $\Omega^{C_{1}-F}$, $\Omega^{C_{2}-F}$, …, $\Omega^{C_{5}-F}$ for the factors. The results are shown in the last column of Table 4, Table 5, Table 6, Table 7, Table 8 and Table 9.

**Step 5**: Calculate the consistency index using (Equation 2) for $A^{G-C}$, $A^{C_{1}-F}$, $A^{C_{2}-F}$, …, $A^{C_{5}-F}$. The results are shown in Table 10. All pairwise comparison matrices meet the $C_{R}<0.1$ requirement.

**Figure 10 entropy-27-00897-f010:**
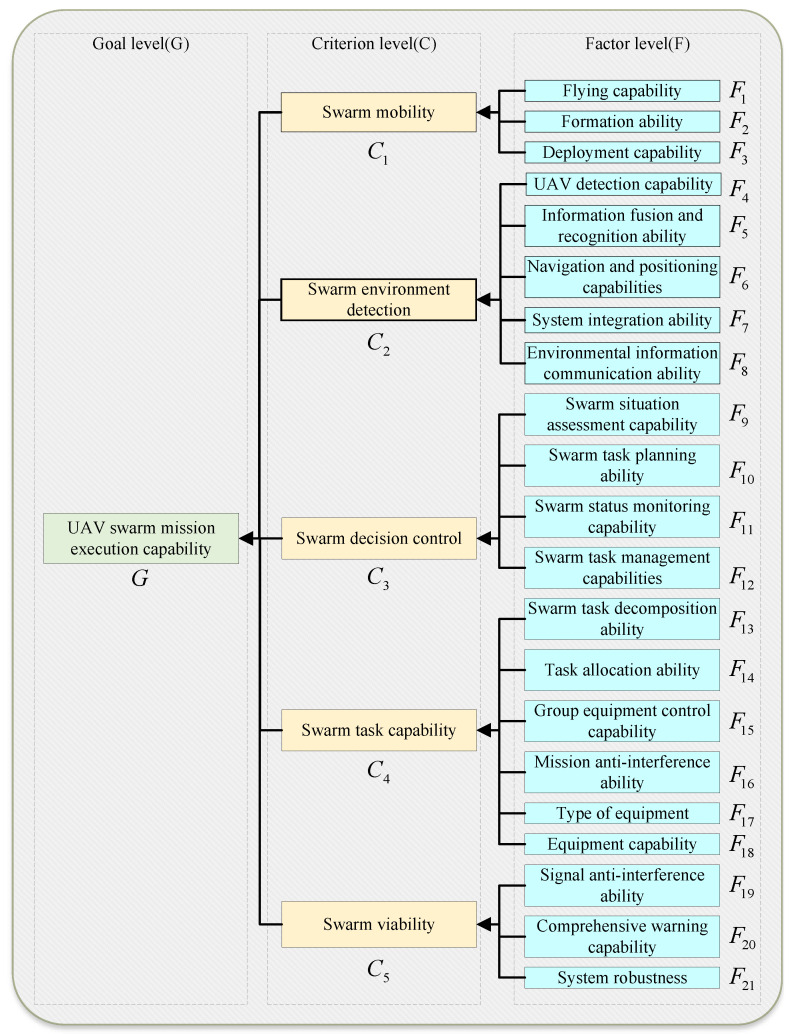
Hierarchical structure for UAV swarm performance evaluation.

**Table 4 entropy-27-00897-t004:** Pair-wise comparison matrix of view (*G*−*C*) and $\Omega^{G-C}$.

*G*	$C_{1}$	$C_{2}$	$C_{3}$	$C_{4}$	$C_{5}$	$\Omega^{G-C}$
$C_{1}$	1	0.4	0.6	0.8	1.5	0.143
$C_{2}$	2.5	1	1.2	1.5	3	0.309
$C_{3}$	1.667	0.833	1	1.5	3	0.265
$C_{4}$	1.25	0.667	0.667	1	2	0.188
$C_{5}$	0.667	0.333	0.333	0.5	1	0.095

**Table 5 entropy-27-00897-t005:** Pair-wise comparison matrix of view ($C_{1}$−*F*) and $\Omega^{C_{1}-F}$.

$C_{1}$	$F_{1}$	$F_{2}$	$F_{3}$	$\Omega^{C_{1}-F}$
$F_{1}$	1	1	0.5	0.261
$F_{2}$	1	1	1.2	0.349
$F_{3}$	2	0.833333	1	0.390

**Table 6 entropy-27-00897-t006:** Pair-wise comparison matrix of view ($C_{2}$−*F*) and $\Omega^{C_{2}-F}$.

$C_{2}$	$F_{4}$	$F_{5}$	$F_{6}$	$F_{7}$	$F_{8}$	$\Omega^{C_{2}-F}$
$F_{4}$	1	1.5	2	5	1	0.306
$F_{5}$	0.667	1	2	4	1	0.248
$F_{6}$	0.5	0.5	1	3	0.6	0.152
$F_{7}$	0.2	0.25	0.333	1	0.5	0.068
$F_{8}$	1	1	1.667	2	1	0.226

**Table 7 entropy-27-00897-t007:** Pair-wise comparison matrix of view ($C_{3}$−*F*) and $\Omega^{C_{3}-F}$.

$C_{3}$	$F_{9}$	$F_{10}$	$F_{11}$	$F_{12}$	$\Omega^{C_{3}-F}$
$F_{9}$	1	0.8	0.8	1	0.221
$F_{10}$	1.25	1	0.8	2	0.294
$F_{11}$	1.25	1.25	1	1	0.277
$F_{12}$	1	0.5	1	1	0.208

**Table 8 entropy-27-00897-t008:** Pair-wise comparison matrix of view ($C_{4}$−*F*) and $\Omega^{C_{4}-F}$.

$C_{4}$	$F_{13}$	$F_{14}$	$F_{15}$	$F_{16}$	$F_{17}$	$F_{18}$	$\Omega^{C_{4}-F}$
$F_{13}$	1	1.5	1.5	3	5	2	0.288
$F_{14}$	0.667	1	1.2	2	5	2	0.226
$F_{15}$	0.667	0.833	1	1.2	5	2	0.196
$F_{16}$	0.333	0s.5	0.833	1	5	0.8	0.129
$F_{17}$	0.2	0.2	0.2	0.2	1	0.5	0.043
$F_{18}$	0.5	0.5	0.5	1.25	2	1	0.117

**Table 9 entropy-27-00897-t009:** Pair-wise comparison matrix of view ($C_{5}$−*F*) and $\Omega^{C_{5}-F}$.

$C_{5}$	$F_{19}$	$F_{20}$	$F_{21}$	$\Omega^{C_{5}-F}$
$F_{19}$	1	1.2	1.2	0.374
$F_{20}$	0.833	1	0.8	0.290
$F_{21}$	0.833	1.25	1	0.336

**Table 10 entropy-27-00897-t010:** Consistency ratio $C_{R}$ for *G*, $C_{1}$, $C_{2}$, $C_{3}$, $C_{4}$, $C_{5}$.

Factor	*G*	$C_{1}$	$C_{2}$	$C_{3}$	$C_{4}$	$C_{5}$
$C_{R}$	0.005	0.083	0.050	0.044	0.055	0.018

Finally, we get $\Omega^{G-C}$ and $\Omega^{C_{i}-F},i=1,2,\ldots ,5$.

Only $\Omega^{G-C}$ and $\Omega^{C-F}$ are listed in Table 4, Table 5, Table 6, Table 7, Table 8 and Table 9, but not $\Omega^{G-F}$. Generally, $\Omega^{G-F}$ is obtained by multiplying each $\Omega^{G-C}$ with $\Omega^{C-F}$. For example, $\omega_{G-F_{6}}=\omega_{G-C_{2}}\cdot \omega_{C_{2}-F_{6}}$.

Note that there are two normalization methods for $\Omega^{G-C}$ and $\Omega^{C-F}$, including sum normalization as (Equation 41) and maximum normalization as (Equation 42).(41)$$\begin{array}{c} \omega_{G-C_{i}}=\frac{\omega_{G-C_{i}}}{\sum \omega_{G-C_{i}}} \end{array}$$(42)$$\begin{array}{c} \omega_{G-C_{i}}=\frac{\omega_{G-C_{i}}}{\max(\omega_{G-C_{i}})} \end{array}$$

Because $\Omega^{G-F}$ is calculated via multiplication, it decreases when there are more nodes if the former normalization method is used. This trend can be prevented by the latter. To identify the two procedures, we use $\Omega^{G-F}$ to designate the result obtained by (Equation 41) and ${\Omega^{\prime }}^{G-F}$ to designate the result obtained by (Equation 42). Table 11 lists $\Omega^{C-F}$, $\Omega^{G-F}$, and ${\Omega^{\prime }}^{G-F}$, with the weight in the first row and the rank in the second row. In addition, the results presented in Table 11 are sum normalized for the convenience of comparison.

#### 5.2.2. Phase 2: Initializing the Bayesian Network (M-HBNS)

**Step 6** In BN, the AHP factors *G*, $C_{1}$, …, $C_{5}$, $F_{1}$, …, $F_{20}$, and $F_{21}$ are turned into nodes, where $C_{1}$–$C_{5}$ are the parents of *G*, $F_{1}-F_{3}$ are the parents of $C_{1}$, $F_{4}-F_{8}$ are the parents of $C_{2}$, $F_{9}-F_{12}$ are the parents of $C_{3}$, $F_{13}-F_{18}$ are the parents of $C_{4}$, and $F_{19}$–$F_{21}$ are the parents of $C_{5}$. Each node is binarized. A detailed discussion of these evaluation metrics would require extensive space and risk overshadowing the central contributions of our work—namely the proposed methodology. Therefore, we have provided only a concise overview in the main text, with comprehensive details available in the cited literature.

**Step 7** Define initial marginal probabilities $P(X_{i})$ for factor nodes and P(Y) for outcome nodes, along with the expected direction of influence $o_{i}$ (positive/negative correlation). These initial expert estimates help constrain the solution space for deriving CPTs.

**Step 8** Under the conditions given in Step 7, $\min\left(scope_{i}\right)=0.5$. As a result, $\left(\omega_{1},\omega_{2},\ldots ,\omega_{n}\right)=2\left(s_{1},s_{2},\ldots ,s_{n}\right)$.

**Step 9** Calculate the conditional probability $P(node|par_{i}^{k})$, where node=$\left\{G,C_{1},C_{2},C_{3},C_{4},C_{5}\right\}$, $par_{i}^{k}$ means the *i*-th parent of the *k*-th node.

**Step 10** Construct the BN parameter.

The above process is repeated until all nodes are calculated.

After completing Phase 2, we compare the HBNS and AHP, and the results of HBNS ${S^{\prime }}^{C-F}$ and ${S^{\prime }}^{G-F}$ are also displayed in Table 11.

Table 11 shows the comparison of the three approaches with different goals. The weights of C−F obtained by the HBNS and AHP are almost equal, and the ranking is identical. However, there are some differences in the weight of G−F. To compare the results more intuitively, a bar chart is shown in Figure 11. As mentioned in Phase 1, there are more factors involved in $C_{2}$ and $C_{4}$, which result in $\omega_{G-F_{i}}$ being smaller than ${\omega^{\prime }}_{G-F_{i}},{s^{\prime }}_{G-F_{i}},i\in \left\{4-8,13-18\right\}$. Conversely, $C_{1}$, $C_{5}$ involve fewer factors and $\omega_{G-F_{i}}$ are larger than ${\omega^{\prime }}_{G-F_{i}},{s^{\prime }}_{G-F_{i}},i\in \left\{1-3,19-21\right\}$. However, HBNS overcomes this disadvantage.

While AHP calculates the overall weight of factors relative to the main goal G−F through direct multiplicative aggregation down the hierarchy, HBNS employs VE on the constructed BN to determine the conditional probability $P(F_{i}|G)$ and then uses (Equation 29). This BN-based probabilistic inference accounts for the full network structure and parameterization, offering a more nuanced and arguably more robust aggregation of influence compared to simple hierarchical multiplication. For an engineering manager, this means a more reliable understanding of how low-level system attributes (*F*-nodes) cumulatively impact strategic objectives (*G*-node), leading to better-informed priorities for investment or intervention.

#### 5.2.3. Phases 3 and 4: Simulation for Learning and Re-Evaluation

These phases demonstrate the BNDM framework’s adaptive learning: initial expert-assigned weights are refined using simulated data, and the estimates converge toward the underlying factor importances.

We consider three networks. BN-initial is constructed from expert knowledge in Phase 2 and serves as the starting point. BN-benchmark shares the same hierarchical structure as BN-initial but has different CPT parameters: we perturb the expert-derived weights to produce an alternative set of sensitivities and then back-solve for the CPTs via M-HBNS, yielding a distinct “ground truth.” Synthetic datasets are generated from BN-benchmark. BN-train is initialized as BN-initial and is then iteratively updated with these synthetic observations using MAP. This setup allows us to observe learning dynamics and quantify convergence toward BN-benchmark.

Table 11 shows the weights for BN-initial and BN-benchmark. Note that the weights of BN-benchmark here are for demonstration purposes only and have no practical significance. Additionally, BN-train is identical to BN-initial at the beginning of training. To qualitatively analyze the weights $S^{\prime G-F}$ of BN-initial, BN-benchmark, and BN-train, we adopted the Manhattan distance shown in (Equation 43):(43)$$\begin{array}{c} D\left(net_{1},net_{2}\right)=\sum_{i=1}^{21}\left|F_{i}^{net_{1}}-F_{i}^{net_{2}}\right|,net_{1},net_{2}\in \left\{initial,benchmark,train\right\} \end{array}$$

The experimental setup involves generating data samples from BN-benchmark, reflecting the true underlying relationships. These samples are then used to update the parameters of BN-train through MAP. We set $D\left(initial,benchmark\right)=1.148$.

Figure 12 illustrates the convergence, showing that the Manhattan distance D(train,benchmark) diminishes as the volume of training data increases. With 10,000 samples, the average distance $\bar{D}\left(\mathrm{train},\;\mathrm{benchmark}\right)$ reduced to 0.075. The resulting sensitivity indices ${S^{\prime }}^{G-F}$ for the BN-train network, shown in Table 12, closely align with those of the BN-benchmark. This empirically validates the framework’s crucial ability to correct and refine potentially biased or incomplete initial expert judgments through data-driven learning, a vital feature for robust engineering management decision support.

Moreover, the convergence of BN-train’s weights towards those of BN-benchmark (Table 12 and Figure 12) demonstrates a critical capability for engineering management: the framework can systematically refine initial, potentially flawed, and expert intuition as the objective data accumulates. This ensures that decision support evolves and improves over time, leading to more accurate and reliable performance assessments.

The ${S^{\prime }}^{G-F}$ of BN-benchmark and BN-train is almost equal, which indicates that, although the expert’s intuition is wrong, the ${S^{\prime }}^{G-F}$ can be optimized with data. Figure 13 shows how the ${S^{\prime }}^{G-F}$ of some nodes changes when the sample size is 4000. Figure 13a represents the change in nodes with a relatively large weight, Figure 13b represents the change in all nodes, and Figure 13c represents the change in nodes with a small weight. These results indicate that the weight of each node approaches the weight of the BN-benchmark, which aligns with our expectation and validates the algorithm’s ability to optimize weights based on data.

## 6. Discussion

This section interprets the key findings from our experimental validation, starting with the direct managerial insights derived from the case study’s performance indicators, followed by a discussion of the framework’s methodological contributions, advantages, and future directions.

### 6.1. Managerial Insights from Key Performance Factors

A primary outcome of our framework is the identification of critical performance drivers, which provides direct, actionable insights for engineering managers. These sensitivity estimates are conditional on the expert-specified DAG and CPT initialization; they reflect that model’s assumptions and therefore do not constitute independent empirical validation of factor importance. The sensitivity analysis of our case study model revealed a clear hierarchy of factor importance, highlighting that capabilities related to Swarm Environment Awareness ($C_{2}$) and Swarm Decision Control ($C_{3}$) are paramount for mission success. The specific insights are as follows:

UAV Detection Capability ($F_{4}$), a component of swarm Environment Awareness ($C_{2}$), emerges as a highly influential factor. This underscores the fact that investment in advanced sensing and detection technologies is not merely a technical upgrade but a strategic imperative. Robust detection capabilities are fundamental for mitigating collision risks, enhancing overall situational awareness for informed decision making, and enabling efficient navigation, all of which directly contribute to mission safety, reliability, and overall success. This finding should guide managers in prioritizing resource allocation towards capabilities that significantly bolster environmental perception.

The high sensitivity associated with Information Fusion and Recognition Ability ($F_{5}$), also a key element of swarm Environment Awareness ($C_{2}$), sends a clear message to engineering managers regarding the importance of a swarm’s data processing capabilities. Beyond merely collecting data, the ability to effectively fuse information from diverse sensors and accurately recognize patterns is crucial. For managers, this highlights the need to invest in sophisticated onboard processing and AI-driven analytics to transform raw data into actionable intelligence.

Communication Ability ($F_{8}$), categorized under swarm Environment Awareness ($C_{2}$) yet fundamental to nearly all swarm functions, also ranks as a significant performance driver. For engineering managers, this finding emphasizes that a resilient communication network is the foundational backbone of effective swarm operations. It is essential for sharing situational awareness, coordinating actions, and enabling both collaborative decision making and remote operator intervention. Deficiencies in this area can cascade into fragmented operations, response latency, and mission failure, making investment in robust, secure, and interference-resistant communication systems a strategic priority.

Within the swarm Decision Control ($C_{3}$) criterion, swarm Task Planning Ability ($F_{10}$) emerges as another critical factor. This finding is highly relevant for engineering managers responsible for operational efficiency and mission success. Effective task planning and dynamic re-planning capabilities ensure that resources are optimally assigned and that the swarm can adapt to changing priorities, indicating that investments in advanced scheduling and coordination protocols are likely to yield significant returns.

In summary, the weights highlight the critical importance of situational awareness and decision-making capabilities within the UAV swarm. Factors related to efficient task planning, communication, and situational assessment are prioritized, reflecting their essential roles in maintaining coordination, safety, and mission success. These insights emphasize the need for advanced capabilities in environment awareness and decision control to optimize the performance of UAV swarms in complex and dynamic settings.

Future research should focus on further enhancing these critical capabilities. Developing more sophisticated detection algorithms and improving data fusion techniques will significantly boost situational awareness. Additionally, advancements in communication protocols will enhance real-time coordination and decision making within the swarm. Exploring machine learning approaches for dynamic task planning and situation assessment could lead to more autonomous and resilient UAV swarms. These research directions will contribute to the development of more capable and efficient UAV systems, capable of tackling increasingly complex missions.

### 6.2. Principal Findings

The experimental validation yielded several key findings that confirm the BNDM framework’s design principles and practical utility. The results highlight distinct advantages over traditional MCDM and sensitivity analysis techniques in the following areas:1.Data-efficient sensitivity analysis: The comparative analysis against the SOBOL method (Section V-A) demonstrated that the analytical HBNS method is highly data-efficient. By leveraging the Bayesian Network’s parametric structure and incorporating expert priors, it achieves accurate sensitivity results with substantially fewer samples, presenting a significant practical advantage in resource-constrained environments.2.Robust and Interpretable Evaluation: The case study (Section V-B, Phase 2) shows that our framework yields a more robust and interpretable assessment than traditional approaches. By using probabilistic inference for weight aggregation, the BN avoids the structural biases of AHP’s multiplicative scheme and provides a more faithful account of system interactions. We further adopt CPTs as a unified, operationally interpretable parameterization: expert judgments are encoded as Dirichlet hyperparameters over CPT entries, while empirical evidence enters as multinomial counts, yielding a transparent posterior fusion within a single probabilistic model. Consequently, managers can inspect the model’s logic directly on the graph, and the accompanying sensitivity analysis pinpoints the most influential drivers, making the insights both actionable and trustworthy.3.M-HBNS for CPT Initialization: We provide a practical M-HBNS procedure to derive BN conditional probabilities from expert-assigned weights (pairwise comparisons), creating an interpretable bridge from MCDM outputs to CPT initialization with explicit feasibility constraints. This mechanism also supports principled prior specification via Dirichlet hyperparameters for subsequent Bayesian updating.4.Adaptive Fusion of Expertise and Data: The framework’s ability to create a synergistic fusion of expert knowledge and empirical data was validated throughout the case study (Section V-B, Phase 3–4). The M-HBNS method provides a transparent mechanism to initialize the model from expert judgments, while the subsequent Bayesian learning process allows the framework to be systematically refined as new operational data becomes available. This dual capability yields an adaptive decision model that is robust in data-scarce regimes yet dynamically responsive over time, delivering transparent, lifecycle-wide decision support—from early design and requirements setting, through deployment and operations, to post-mission evaluation—so managers can continuously reassess performance and adjust strategies as new insights emerge.

Our framework fills a critical gap by providing engineering managers with a tool that is not only technically robust but also designed to support iterative strategic evaluation, resource allocation, and risk management in the complex lifecycle of UAV swarm deployment. It moves beyond static, isolated assessments to offer a dynamic and integrated decision-support system.

### 6.3. Limitations and Future Research

Despite the framework’s contributions, this study has several limitations that provide clear avenues for future research. First, the framework’s mathematical tractability currently relies on a hierarchical structure and an assumption of parent–node independence as formalized in Corollary 1. This may not fully capture the complex, non-hierarchical interdependencies present in some real-world systems. Second, the initial model construction is heavily dependent on a resource-intensive expert elicitation process, posing a practical challenge for deployment. Finally, the current validation, while comprehensive, is based on a simulated environment.

These limitations directly inform future research directions. To address the structural constraints, future work should focus on extending the HBNS and M-HBNS methods to handle more general Directed Acyclic Graphs, which would allow for modeling more intricate system interdependencies. The challenge of expert dependency could be mitigated by investigating semi-supervised learning techniques for model construction.

Our choice of an expert-elicited hierarchical BN emphasizes interpretability, stability, and managerial auditability, at the cost of constraining interdependencies to a layered DAG. In data-scarce settings, this design reduces variance and supports accountable decisions, but it may miss cross-criterion links. In contrast, BN structure learning (e.g., score-based or constraint-based algorithms, possibly with order or edge priors) can uncover interdependencies when sufficient data are available, albeit with higher data demands and potentially reduced transparency. Importantly, the proposed HBNS/M-HBNS methods operate on any DAG; thus, once additional data accrue, the hierarchy can be relaxed and the structure learned or refined, after which HBNS can still deliver sensitivity readouts. Exploring this generalization and quantifying the accuracy–interpretability–data trade-off is a key avenue for future work.

In the case study, we binarize nodes to leverage the closed-form M-HBNS mapping and to keep results interpretable; HBNS itself remains applicable to multi-state nodes, and extending the inverse step via the above constrained formulation is a promising direction for future work.

A crucial next step is to enhance the framework’s real-time performance for tactical decision making, potentially through model approximation, more efficient inference algorithms, or hardware acceleration.

Ultimately, the most vital future work involves applying the BNDM framework to real-world operational data from specific scenarios. For instance, in logistics, it could quantify the trade-off between delivery speed and operational cost; in disaster response, it could optimize swarm configurations for robust communication in degraded environments. Applying the BNDM framework to these real-world contexts would not only validate its effectiveness but also help develop domain-specific evaluation benchmarks, representing a vital next step in this research.

## 7. Conclusions

In this study, we developed and validated a novel Bayesian Network-based Multicriteria Decision-Making (BNDM) framework to address the complex challenge of UAV swarm performance evaluation. The core contribution of this work is a theoretically grounded, bidirectional methodology that uses variance decomposition to unify the qualitative weights of expert-driven methods like AHP with the quantitative, probabilistic parameters of a Bayesian Network. Our comprehensive experimental validation confirmed that this framework is not only accurate and highly data-efficient but also an adaptive learning system capable of refining initial expert judgments as new empirical data becomes available.

For engineering managers, this framework provides a robust, transparent, and dynamic decision-support tool. It moves beyond the limitations of static or “black-box” approaches, enabling a more agile and evidence-based style of management. The analysis from our case study yielded direct, actionable insights, identifying situational awareness and coordinated decision making as the most critical drivers of mission success. This empowers managers to optimize resource allocation, mitigate operational risks, and enhance overall mission effectiveness with greater confidence and strategic foresight.

This work represents a significant step towards the more scientific management of complex autonomous systems. As swarm technologies become further integrated into critical commercial and civil operations, frameworks that systematically unify human expertise with data-driven insights will be indispensable. Ultimately, this research offers a clear pathway for organizations to harness the full potential of UAV swarm technology, not merely as a technical capability, but as a strategically managed asset. 

## Figures and Tables

**Figure 1 entropy-27-00897-f001:**
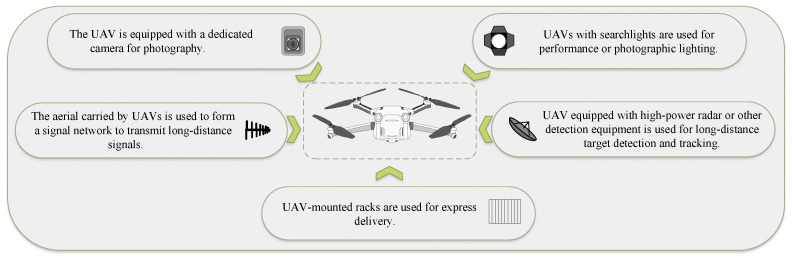
Typical configurations of UAVs with different functions: illustrates UAVs equipped for photography, illumination, express delivery, long-distance communication, and target detection.

**Figure 2 entropy-27-00897-f002:**
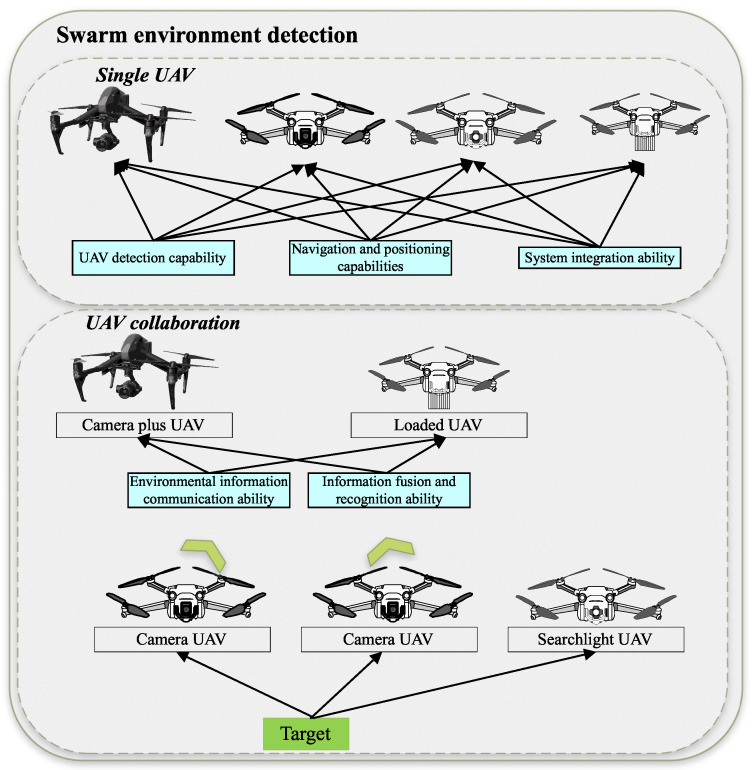
Environmental detection in UAV swarms: Depicts the collaborative sensing and communication mechanisms that enable real-time environmental awareness within UAV swarms.

**Figure 3 entropy-27-00897-f003:**
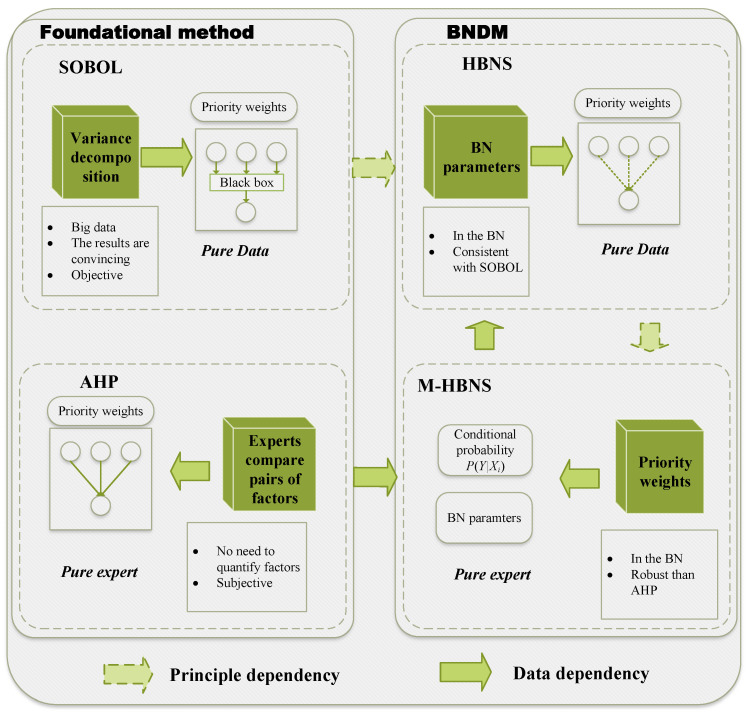
Dependencies between the core methods in the proposed BNDM framework, illustrating the flow of information from pure expert-based methods (AHP) to data-driven techniques (SOBOL and BN). The integration of these approaches results in the Bayesian decision-making model.

**Figure 4 entropy-27-00897-f004:**
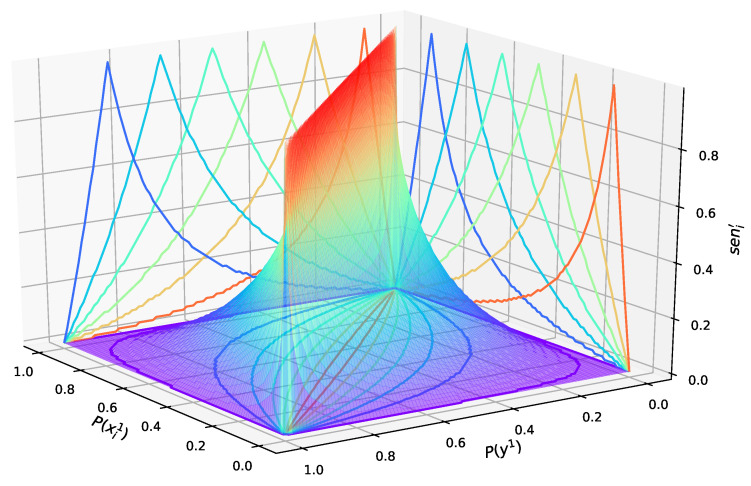
Feasible solution space for initializing Bayesian Network parameters $P(x_{i}^{1})$ and $P(y^{1})$ given a target sensitivity $s_{i}^{\prime }$, based on Equation (Equation 35). This illustrates the interdependent constraints guiding the M-HBNS method when converting expert-defined weights into CPTs.

**Figure 5 entropy-27-00897-f005:**
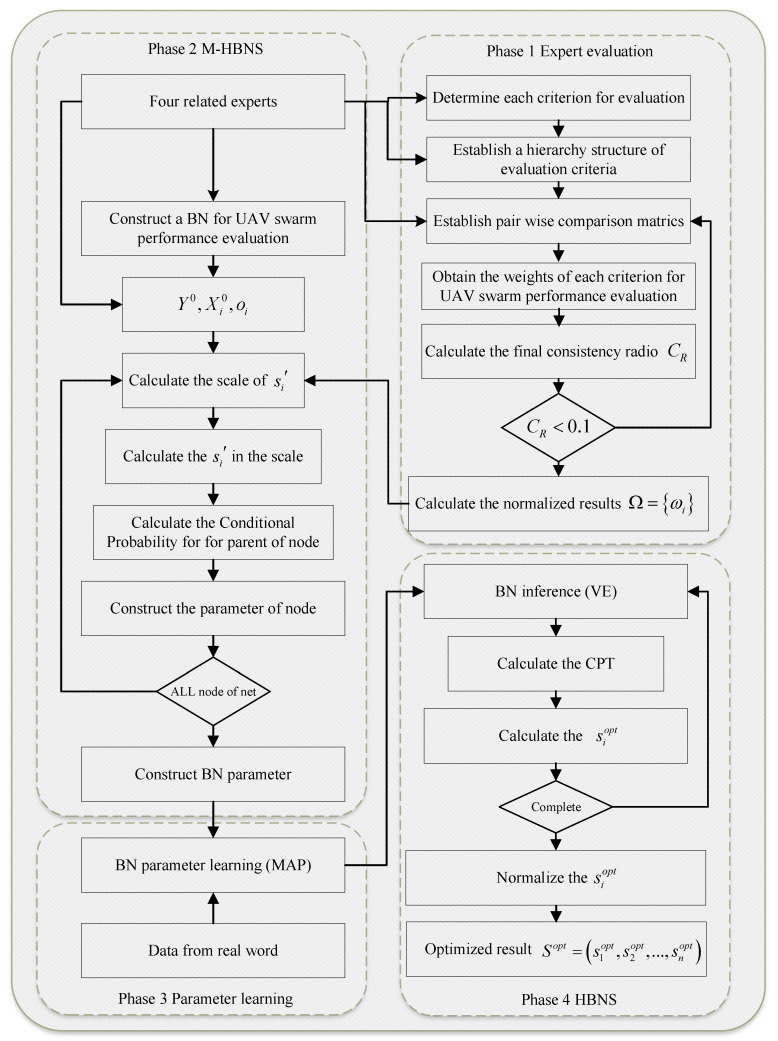
Framework of BNDM.

**Figure 6 entropy-27-00897-f006:**
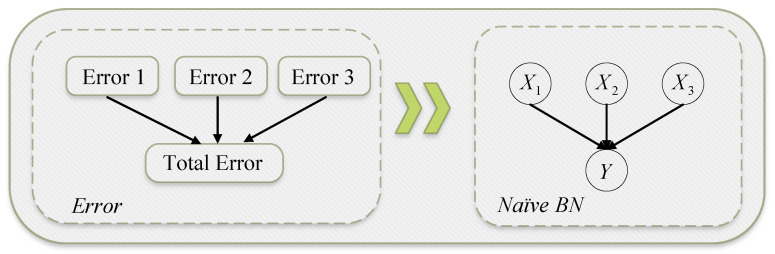
Example of a Bayesian Network.

**Figure 7 entropy-27-00897-f007:**
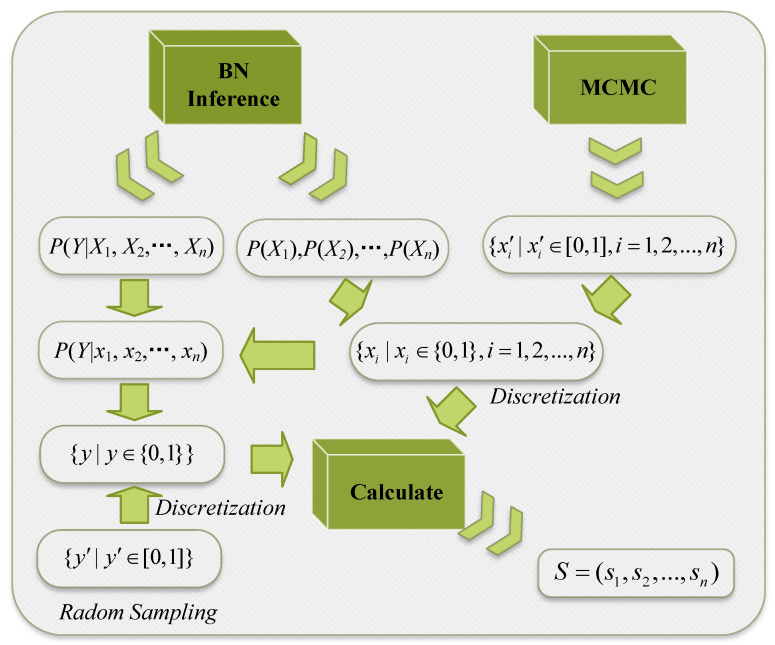
Workflow for generating synthetic discrete samples from the BN in Figure 4 (CPTs in Table 3) and applying SOBOL. HBNS indices are computed analytically on the same BN for a fair comparison.

**Figure 8 entropy-27-00897-f008:**
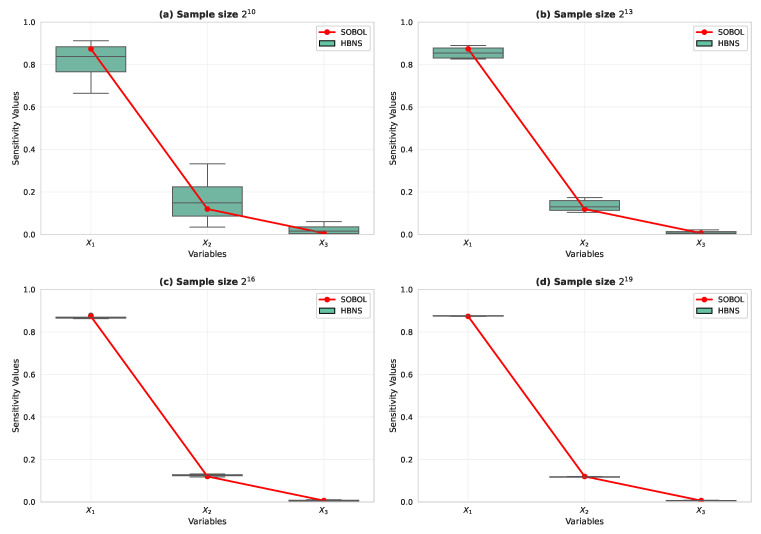
SOBOL vs. HBNS sensitivity indices for $X_{1},X_{2},X_{3}$. Data are synthetic, generated from the BN in Figure 4 with Table 2 CPTs. Boxplots show 10 repetitions for each $N\in \{2^{10},2^{13},2^{16},2^{19}\}$. HBNS uses the analytical formula; SOBOL uses the sampled binary dataset from the same generator.

**Figure 9 entropy-27-00897-f009:**
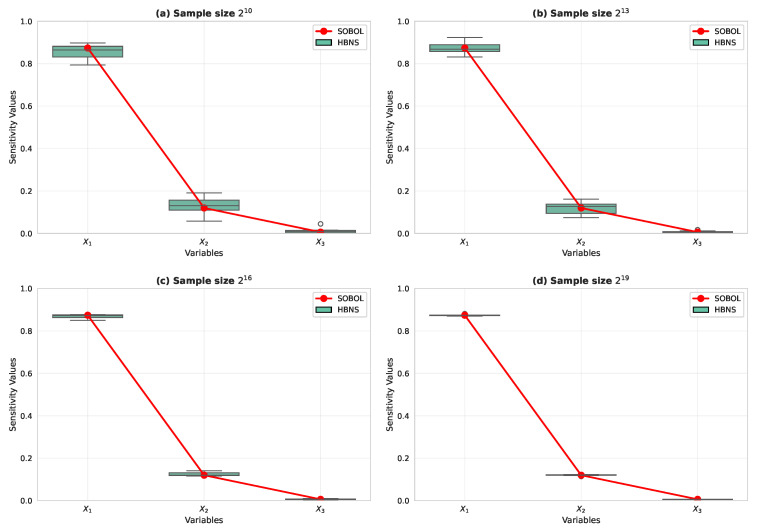
HBNS sensitivity indices after parameter learning from a small synthetic dataset $\left(N=2^{10}\right)$ generated from the BN in Figure 4. Baseline (analytical without learning) vs. post-learning HBNS shows stabilization with limited data when expert-informed priors are used (Equation (Equation 29)).

**Figure 11 entropy-27-00897-f011:**
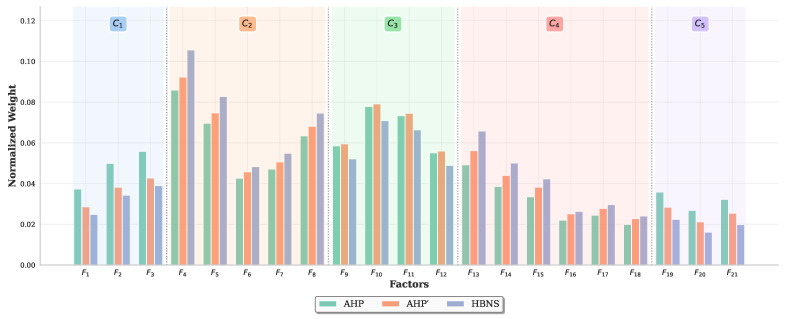
Bar chart comparing global factor weights G−F derived from AHP, AHP’, and HBNS for the UAV swarm example.

**Figure 12 entropy-27-00897-f012:**
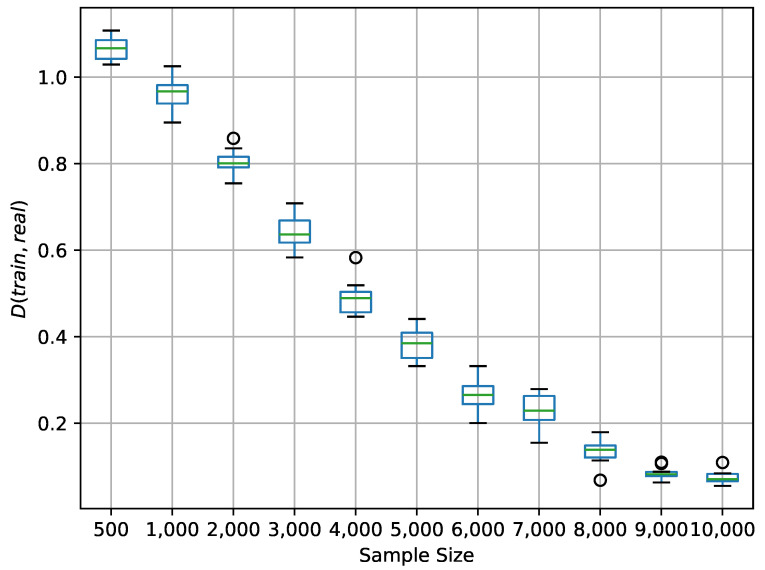
Convergence of **BN-train** toward **BN-benchmark** as the training **synthetic** sample size increases. The y axis is the Manhattan distance between the ${S^{\prime }}^{G-F}$ vectors of BN-train and BN-benchmark. Datasets are generated from BN-benchmark; BN-train is updated via MAP.

**Figure 13 entropy-27-00897-f013:**
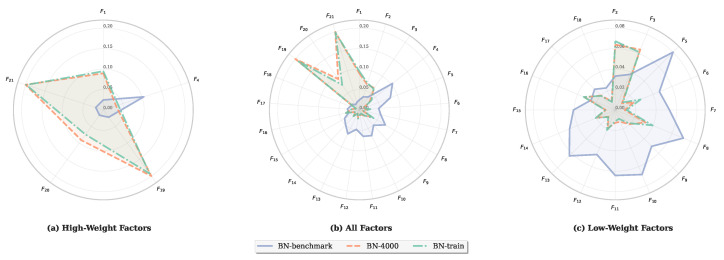
Radar plots of ${S^{\prime }}^{G-F}$ showing convergence under **synthetic** data generated by the BN-benchmark: (**a**) high-weight factors, (**b**) all factors, and (**c**) low-weight factors. BN-train parameters are updated via MAP using Equation (Equation 11).

**Table 1 entropy-27-00897-t001:** Comparison of different MCDM methods.

Reference	Objective Weighting Method	Subjective Weighting Method	Single Decision Maker	Group of Decision Makers	One Pipeline	Interpretable
Nila and Roy [36]	✔	✔	✔	✔		✔
Wang et al. [10]	✔	✔	✔	✔	✔	
Aytekin et al. [37]	✔	✔	✔	✔		
Dehshiri et al. [38]		✔	✔	✔		✔
Nguyen [39]		✔		✔		✔
Alves et al. [32]		✔		✔		✔
Maghsoodi et al. [40]	✔				✔	
Nessari et al. [41]	✔				✔	✔
Lin et al. [33]	✔	✔	✔	✔		✔
Susmaga et al. [34]	✔				✔	✔
Yan et al. [9]		✔		✔	✔	✔
Dahooie et al. [42]	✔				✔	✔
**Ours**	✔	✔	✔	✔	✔	✔

**Table 2 entropy-27-00897-t002:** Values of γ.

n	1	2	3	4	5	6	7	8	9	10	11
γ	0	0	0.58	0.90	1.12	1.24	1.32	1.41	1.45	1.49	1.51

**Table 3 entropy-27-00897-t003:** Parameters of BN.

Name	Size	CPT
$X_{1}$	2	0.2 0.8
$X_{2}$	2	0.3 0.7
$X_{3}$	2	0.4 0.6
*Y*	2	[0.46, 0.43, 0.33, 0.31, 0.78, 0.75, 0.67, 0.65],
		[0.54, 0.57, 0.67, 0.69, 0.22, 0.25, 0.33, 0.35]

**Table 11 entropy-27-00897-t011:** Comparison of normalized factor weights ($\Omega^{C-F}$, $\Omega^{G-F}$, ${\Omega^{\prime }}^{G-F}$) from AHP and sensitivity indices (${S^{\prime }}^{C-F}$, ${S^{\prime }}^{G-F}$) from HBNS for the UAV swarm assessment. Ranks are shown below weights.

*G*	*G*																				
C	$C_{1}$			$C_{2}$					$C_{3}$				$C_{4}$						$C_{5}$		
F	$F_{1}$	$F_{2}$	$F_{3}$	$F_{4}$	$F_{5}$	$F_{6}$	$F_{7}$	$F_{8}$	$F_{9}$	$F_{10}$	$F_{11}$	$F_{12}$	$F_{13}$	$F_{14}$	$F_{15}$	$F_{16}$	$F_{17}$	$F_{18}$	$F_{19}$	$F_{20}$	$F_{21}$
${S^{\prime }}^{C-F}$	0.25	0.35	0.40	0.29	0.23	0.13	0.1	0.20	0.22	0.30	0.28	0.20	0.28	0.21	0.18	0.11	0.13	0.10	0.38	0.28	0.34
	3	2	1	1	2	5	4	3	3	1	2	4	1	2	3	5	4	6	1	3	2
$\Omega^{C-F}$	0.26	0.35	0.39	0.28	0.23	0.14	0.15	0.21	0.22	0.29	0.28	0.21	0.26	0.21	0.18	0.12	0.13	0.11	0.38	0.28	0.34
	3	2	1	1	2	5	4	3	3	1	2	4	1	2	3	5	4	6	1	3	2
${S^{\prime }}^{G-F}$	0.025	0.034	0.039	0.108	0.085	0.049	0.055	0.076	0.051	0.070	0.065	0.048	0.066	0.049	0.042	0.027	0.030	0.024	0.022	0.016	0.020
	17	14	13	1	2	10	7	3	8	4	6	11	5	9	12	16	15	18	19	21	20
$\Omega^{G-F}$	0.037	0.050	0.056	0.086	0.070	0.043	0.047	0.063	0.059	0.078	0.073	0.055	0.049	0.039	0.034	0.022	0.024	0.020	0.036	0.027	0.032
	14	9	7	1	4	12	11	5	6	2	3	8	10	13	16	20	19	21	15	18	17
${\Omega^{\prime }}^{G-F}$	0.029	0.038	0.043	0.092	0.075	0.046	0.051	0.068	0.060	0.079	0.075	0.056	0.056	0.044	0.038	0.025	0.028	0.023	0.028	0.021	0.025
	15	13	12	1	3	10	9	5	6	2	4	8	7	11	14	19	17	20	16	21	18

**Table 12 entropy-27-00897-t012:** ${S^{\prime }}^{G-F}$ sensitivity indices for **BN-initial** (expert-initialized), **BN-benchmark** (synthetic data generator), and **BN-train** after MAP updates using **10,000 synthetic samples** drawn from the BN-benchmark. This illustrates how data refine expert-initialized weights.

Network	$F_{1}$	$F_{2}$	$F_{3}$	$F_{4}$	$F_{5}$	$F_{6}$	$F_{7}$	$F_{8}$	$F_{9}$	$F_{10}$	$F_{11}$	$F_{12}$	$F_{13}$	$F_{14}$	$F_{15}$	$F_{16}$	$F_{17}$	$F_{18}$	$F_{19}$	$F_{20}$	$F_{21}$
Initial	0.025	0.034	0.039	0.106	0.083	0.048	0.055	0.075	0.052	0.071	0.066	0.049	0.066	0.050	0.042	0.026	0.030	0.024	0.022	0.016	0.020
Benchmark	0.091	0.066	0.066	0.033	0.010	0.020	0.010	0.033	0.020	0.013	0.013	0.020	0.010	0.021	0.010	0.034	0.021	0.010	0.203	0.093	0.203
Train	0.096	0.070	0.063	0.039	0.011	0.028	0.012	0.041	0.019	0.010	0.011	0.022	0.010	0.021	0.010	0.035	0.020	0.009	0.196	0.076	0.202

## Data Availability

Data are contained within the article. The source code is available at https://github.com/phonixer/Bayesian-Network-Decision-Making-A-Unified-Bayesian-Framework (accessed on 6 July 2025).

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
