# Peer review of "Bridging Intuition and Data: A Unified Bayesian Framework for Optimizing Unmanned Aerial Vehicle Swarm Performance"

_entropy, 2025, doi:10.3390/e27090897_

Round 1

Reviewer 1 Report

Comments and Suggestions for Authors

The paper provides a Bayesian Network (BN) based framework to modelling swarm performance based on a strict hierarchical structure of Factors → Criteria → Performance Goal. The key aim is to identify the factors which most affect UAV swarm performance. The proposed framework has four steps:

  1. Use the established Analytic Hierarchy Process (AHP) to ask domain experts to specify the hierarchical structure (which factors and criteria to use) and pairwise comparisons between factors. These pairwise comparisons are then used to compute weights which reflect the importance of factors to criteria and hence to the overall goal.
  2. A novel approach, M-HBNS, which translates these weights into Conditional Probability Tables (CPT) parameters in the BN.
  3. Refine those expert-defined parameters by treating them as priors and modifying them with experimental results - to give CPTs which combine expert-elicitation and data-driven values
  4. Use the HBNS approach to convert these learnt CPTs into SOBOL sensitivities which can be interpreted by practitioners (e.g. swarm managers) ito understand factor importance.

Detailed Comments

  • Introduction

Line 43: You mention that managing UAV swarms is difficult because of emergent and non-linear behaviour. Could you comment - perhaps in the Discussion - the extent to which your approach addresses this, and how it does so.

Line 49: Please be a little more specific about which modern machine learning-based methods are "black-box". Not all ML methods are opaque, for example decision trees aren't. I guess you are thinking mostly about neural networks? Please support with a reference.

Line 58: Wang et al [10] is a very specific reference describing a specific BN structure learning algorithm. A more general reference covering BNs might be more useful e.g. Daphne Koller and Nir Friedman. Probabilistic graphical models: principles and techniques. MIT press, 2009.

  • Literature Reviews

Line 184: Please give some examples of mathematical models used (entropy weighting? PCA?) and some references.

  • Preliminaries

Line 289: "edges represent correlations" is misleading here, because variables in a BN may be correlated even if there is no edge between them. It may be more accurate to say that an edge means that one node directly influences the other, or put another way they are NOT conditionally independent.

Line 292: once again the Wang et al [10] is quite limited, and it would be good to expand a little on what structure learning, knowledge elicitation and parameter specification means in the context of BNs. This also gives you an opportunity to stress that you are providing an alternative, and novel way of specifying parameters - M-HBNS

Line 307: did you write the code yourself to do variable elimination? Is it available? Or did you use a package to do this?

  • Model Construction

Line 322: In my opinion, the paper provides only a limited exploration of the use of BNs in this application domain. For example, a simple hierarchical structure is assumed, and experts specify which factors are associated with which criteria and estimate the weights. One of the strengths of BNs is their ability to explore interdependencies between variables. A discussion of the tradeoffs between your approach and for example BN structure learning would be valuable.

Line 324: a diagram illustrating the hierarchical BN would be helpful here

Line 326: "single root node" is very misleading here, because in BNs/DAGs, a root node is one with no parents, that is, the factors in your model.

Line 330: Similarly in a BN/DAG context a leaf node is one with no children - in your case the Goal node.

Line 361: The term ‘naive BN’ may be confusing here, as it risks conflation with the Naive Bayes model which has one node with many children, but here you have one node (Y) with many parents. So it might be safer to use "simple BN" instead.

Line 393: I think you need to be a little clearer here that you have used binary variables in the model evaluation, but many of your results (e.g. equation 28) apply to categorical variables with more than two states.

Line 404: Please be clearer about what scope is, and how it is computed/chosen.

  • Experimental evaluation

Line 604: I think you should describe the process by which the hierarchical structure was specified by four domain experts, informed by a literature review, in much more detail. What papers were consulted in the review, and how disagreements between experts were resolved. This would be useful for anyone trying to replicate your approach.

Line 633: "Each node is binarized". Could you please say more about why this was done (since some your theoretical results relate to variables with more than two states), and how it was done. Binarization can lead to loss of information, arbitrary thresholding, and reduced sensitivity resolution.

The paper would benefit from a discussion of these trade-offs, including how the binarization was performed, whether alternative discretizations were considered, and how this choice may have influenced the results. The paper might benefit from some sensitivity studies investigating alternative approaches.

Line 668: Please provide more details of how the BN-benchmark used to create training data was specified. Was it specified by experts? How does it relate to BN-initial? Does it have the same structure but different parameters? 

Figure 10: I believe that this shows that increasing the amount of training data (generated from the benchmark network) makes the sensitivity values move closer to those of the benchmark network. This usefully illustrates Bayesian updating but dean't seem to be a very novel contribution from the perspective of BN learning. This aspect could be strengthened  by exploring the impact of different prior strengths, modeling uncertainty in expert input, or quantifying the influence of data on posterior distributions. 

  • Discussion

Line 707: "capabilities related to …. are paramount" - since the BN structure and dependencies were defined a priori through expert elicitation and literature review, the identified ‘important’ factors are therefore largely a reflection of those initial assumptions, rather than an independent empirical discovery. It would be helpful for the authors to clarify that the sensitivity results are conditional on the expert-defined model, and do not constitute independent empirical validation of factor importance.

Line 749: "Principal Findings". I don't think you emphasise enough the contribution you make with the M-HBNS method. This approach of deriving weights from pairwise factor comparisons could be of value to the BN community as a whole. Therefore, you might add a new bullet point emphasising its contribution.

Line 762: "the graphical nature of the BN offers greater transparency". I don't think this has been demonstrated very clearly, the process produces sensitivity values, but does illuminate the causal pathways very clearly. Perhaps you could discuss more how this has or could be achieved.

Line 765. "Adaptive Fusion of Expertise and Data". The paper has illustrated that weights converge towards empirical value as sample size increases, but this is a well-established behaviour in Bayesian approaches and therefore may not be a novel contribution. You could perhaps strengthen this aspect by discussing how in practice you can balance the expert-specified weights with the data-derived influence. 

Minor Points

Line 47: It would be good to explain what MCDM stands for here. Although this is explained in the keywords, this is its first use in the main text.

Line 96: It might be good to make "Swarm Mobility" etc bold text just to make these stand out a little more.

Table 1: "Interpreter" column should be "Interpretable"

Line 247: A space required after "level"

Line 350: This is the first time you have used HBNS and M-HBNS in the text so maybe you need to explain what they stand for.

Line 446: "bounded by" is perhaps clearer than "composed of" to describe the volume encompassed by these surfaces.

Line 486: please clarify how scope is calculated.

Figure 6: I would suggest it might be better if the subplots in Figure 6 have exactly the same vertical axes so that it is clearer that the HBNS plot is the same at each sample size. 

Table 4: would be clearer if this appeared before Tables 5 - 9.

Line 649: "identity" should be "identical"

Figure 9: it would be helpful to identify what C1, C2 represent e.g. "swarm mobility" so that users can more easily see the importance of the five criteria on this figure.

Reviewer 2 Report

Comments and Suggestions for Authors

Dear Authors,

I recommend further clarifying your citation system to clearly distinguish between equations drawn from the literature and those newly proposed in your work.

In Theorem 1, you mention "causality independent." Could you please specify where and how you checked for causality within your model?

Please enhance the legends for all figures and tables. The legends should provide a complete and self-contained description, even if some information is already presented in the main text.

Most standard descriptions of BN (Bayesian Networks) are associated with a specific dataset, yet your example does not reference any data. Please include a comprehensive dataset description. Additionally, some papers recover the CPT (Conditional Probability Table) without relying on data. How might knowledge of this kind be incorporated into your approach?

Finally, I suggest including a simulated data example to clearly demonstrate the limitations and strengths of your proposed method.

Round 2

Reviewer 1 Report

Comments and Suggestions for Authors

I am happy with all the revisions made with the exception of the following two minor points:

  1. Literature Reviews

Revision 4, Line 186: The new sentence added at line 186 didn't make much sense, especially as it would have been better to include it after the sentence beginning "Objective weighting techniques …" to which it refers. Please move this revision and make it clearer.

  1. Model Construction

Revision 9, Figure 8: Thank you for your new Figure 8 which is very clear, but which I would suggest is missing one key detail. A key feature of a Bayesian Network is that is is a directed graph, so I think it would be helpful if you placed some arrows on the figure to illustrate the causal flow - that is, the Factors cause the Criterion assessment (arrows from factors to criteria), and the Criteria assessments cause the overall Goal (arrows from criteria to goal).

Reviewer 2 Report

Comments and Suggestions for Authors

Dear authors

I appreciate the review of the work.

I would like to see a more detailed description of the "Interpretable" model, as it is included in the column of Table 1.

Also, I think it would be great if interested readers could have access to this code for their work. Do you intend to put your work in GitHub or any other repository? Please include the link to it.
